# Active Sampling for Ultra-Low-Bit-Rate Video Compression via Conditional Controlled Diffusion

## Abstract

Diffusion models provide a powerful generative prior for perceptual reconstruction at ultra-low bitrates, but effective video compression requires controlling the generative process using highly compact conditioning signals. In this work, we present **ActDiff-VC**, a diffusion-based video compression framework for the ultra-low-bitrate regime. Our method partitions videos into variable-length segments, transmits keyframes only when needed, and summarizes temporal dynamics using a compact set of tracked point trajectories. Conditioned on these sparse signals, a conditional diffusion decoder synthesizes the remaining frames, enabling perceptually realistic reconstruction under severe rate constraints. To support this design, we introduce two mechanisms: content-adaptive keyframe selection and budget-aware sparse trajectory selection, which together enable compact yet effective conditioning for generative reconstruction. ActDiff-VC has an asymmetric computational profile: on a single NVIDIA A100 GPU, encoding requires 109 ms per frame, while the adopted 20-step diffusion decoder requires 2311 ms per frame, making the framework particularly suitable for ultra-low-bitrate applications such as cloud-assisted reconstruction, archival storage, and offline content distribution, where lightweight encoding and perceptual reconstruction quality are prioritized.

Experiments on the UVG and MCL-JCV benchmarks show that ActDiff-VC achieves up to 64.6% bitrate reduction at matched NIQE, improves KID by up to 64.6% and FID by up to 37.7% at comparable bitrates against strong learned codecs, and delivers favorable perceptual rate–distortion trade-offs relative to learned and generative baselines in the ultra-low-bitrate regime.

## 1 Introduction

As video increasingly underpins everyday communication and entertainment, it has become a dominant contributor to network traffic and storage demand. This scale makes improvements in compression efficiency essential, especially as expectations for visual quality continue to rise while bandwidth budgets continue to shrink. The central challenge is to deliver higher perceived quality under tighter rate constraints. Learned video compression has advanced primarily through distortion-minimizing training, but these principles strain in the ultra-low-bitrate regime. Here, we use ultra-low bitrate to refer to operating points roughly at or below 0.05 bits per pixel, where the transmitted information per frame is severely constrained. At such rates, the encoder can no longer afford to transmit rich motion side information or detailed residual corrections, and reconstructions often become over-smoothed. In this regime, perceived quality is governed more by realism than by pixel-level fidelity, motivating decoders that rely on powerful generative priors to infer missing content from minimal side information.

Diffusion models provide a powerful generative prior for reconstruction at ultra-low bitrates, enabling perceptually realistic synthesis from highly compressed conditioning signals. However, integrating diffusion into video compression raises practical challenges in controlling the generative process under tight rate budgets. Rich conditioning can improve reconstruction fidelity but increases side-information cost, while overly sparse conditioning can lead to perceptually implausible or unstable reconstructions. Existing diffusion-based codecs address this trade-off in different ways. I²VC Liu et al. (2024) leverages latent diffusion priors but performs iterative denoising independently for each frame, limiting efficiency. EVC-PDM Li et al. (2024a)

evaluates candidate reconstructions by running diffusion at the encoder to determine when to transmit additional keyframes, incurring significant encoder overhead. These methods highlight the central challenge of designing compact yet informative conditioning mechanisms that enable high perceptual reconstruction quality in ultra-low-bitrate diffusion-based video compression.

To address this conditioning efficiency challenge, we introduce a generative compression strategy that exploits temporal redundancy through structured sparse conditioning. Instead of reconstructing frames via per-frame diffusion reconstruction, we partition the video into variable-length segments and transmit keyframes when the current keyframe is no longer sufficiently informative for reconstructing subsequent frames. Within each segment, temporal dynamics are summarized by a compact set of tracked point trajectories that serve as motion conditioning signals for generative reconstruction. A conditional diffusion decoder then synthesizes the remaining frames from these sparse signals, enabling perceptually realistic reconstruction at ultra-low bitrates. Our approach is motivated by the observation that motion in natural videos exhibits strong spatial correlation, allowing sparse trajectory conditioning to approximate dense motion fields with minimal perceptual degradation. This enables a compression design in which a small number of transmitted keyframes and compact motion cues are sufficient to guide generative reconstruction of intermediate content. This design maintains lightweight encoding while allocating computation to the diffusion decoder, making ActDiff-VC suitable for cloud-assisted reconstruction, archival storage, and offline content distribution at ultra-low bitrates. Building on this principle, we leverage pre-trained diffusion models to realize sparse motion-guided video reconstruction in the ultra-low bitrate regime, and summarize our contributions as follows:

- We introduce a generative video compression framework in which a conditional diffusion decoder reconstructs video from *sparse trajectory conditioning*, enabling perceptually realistic reconstruction at ultra-low bitrates.

- We propose two encoder-side mechanisms for compact and informative conditioning: (i) *content-adaptive keyframe selection* that forms variable-length segments aligned with scene dynamics, and (ii) a *budget-aware sparse trajectory selection* strategy that extracts representative motion cues from dense tracking to minimize side-information rate.

- We demonstrate strong perceptual compression performance in the ultra-low-bitrate regime on the UVG and MCL-JCV benchmarks, achieving up to 64.6% bitrate reduction at matched NIQE, improving KID by up to 64.6% and FID by up to 37.7% at comparable bitrates against strong learned codecs. We further show favorable perceptual rate–distortion trade-offs relative to learned codecs and generative baselines, and extensive ablations validate each component's contribution.

## 2 Related work

Learning-based video codecs have achieved substantial gains over traditional standards by replacing hand-crafted modules with neural networks (Wiegand et al., 2003; Sullivan et al., 2012). Most early approaches follow a predictive residual-coding paradigm, where motion-compensated predictions are formed from previously decoded frames and the encoder transmits compact residual representations (Lu et al., 2019). Subsequent methods extend this framework through conditional coding (Li et al., 2021; 2022; 2023), which models an explicit distribution of the current frame conditioned on decoded history and entropy-codes it under the learned distribution. A complementary direction is interpolation-based compression (Wu et al., 2018; Alexandre et al., 2023), which synthesizes intermediate frames from sparsely transmitted reference frames. While these learned codecs achieve strong distortion performance at moderate bitrates, they are typically trained with per-frame distortion objectives that lead to over-smoothed and perceptually blurry reconstructions in the ultra-low-bitrate regime (Blau & Michaeli, 2019). This limitation motivates compression strategies that prioritize perceptual realism over strict pixel fidelity under extreme rate constraints.

Perceptual video compression improves visual quality by augmenting learned codecs with perceptual losses and adversarial training, thereby biasing reconstructions toward human visual preferences rather than exact pixel fidelity (Xu et al., 2024). GAN-based methods further enhance high-frequency details and textures through adversarial objectives (Veerabadran et al., 2020; Du et al., 2022; 2024). However, existing perceptual

and GAN-based codecs do not fully address the ultra-low-bitrate regime (e.g., $\leq 0.05$ bpp). In this range, the bit budget severely limits the information that can be transmitted per frame, and distortion-optimized learned codecs often produce noticeably blurry reconstructions. More critically, many perceptual and GAN-based codecs determine their operating points through training-time choices, such as rate–distortion loss weights or separately trained rate models, making them difficult to adapt reliably to previously unseen ultra-low-bitrate settings without retraining. This gap motivates leveraging stronger generative priors that can reconstruct realistic video from highly compact conditioning signals.

Diffusion models (Ho et al., 2020) have emerged as powerful generative priors capable of synthesizing photorealistic content from compact conditioning signals, including text (Nichol et al., 2021; Ho et al., 2022), sketches (Zhang et al., 2023), and point trajectories (Gu et al., 2025). This capability makes them well suited for ultra-low-rate video compression, where the bit budget for transmitted side information is severely constrained. In the image domain, diffusion-based codecs have demonstrated substantial perceptual gains at very low bitrates (Pan et al., 2022; Yang & Mandt, 2023; Lei et al., 2023; Careil et al., 2023; Relic et al., 2024), indicating that diffusion priors can improve reconstruction quality under extreme rate constraints. Extending these benefits to video, however, remains challenging, as reconstruction must both exploit temporal redundancy and operate under highly compressed conditioning.

Recent methods have begun incorporating diffusion priors into learned video compression. I$^2$VC (Liu et al., 2024) encodes frames into latent representations, allocates bits using a learned spatio–temporal importance mask, and reconstructs them with a pre-trained image latent diffusion model conditioned on features from neighboring decoded frames. However, without an explicit motion signal, maintaining consistent reconstruction under large motion or occlusion may be challenging. EVC-PDM (Li et al., 2024a) instead transmits a subset of intra frames and synthesizes skipped frames with a video diffusion model. It selects keyframes by performing diffusion-based forecasting at the encoder and inserting a keyframe when predicted perceptual quality drops below a threshold. However, its keyframe-selection procedure relies on repeatedly running diffusion-based forecasts at the encoder to decide when additional intra frames are required, incurring substantial encoder-side overhead and limiting practicality. These limitations motivate approaches that avoid repeated diffusion-based forecasting at the encoder and instead use compact sparse trajectories as direct conditioning for diffusion-based reconstruction.

Adaptive keyframe placement addresses temporal variation by selecting content-dependent segments instead of relying on fixed group-of-pictures (GOP) structures. Most learned video codecs use a fixed GOP, which can be suboptimal when motion intensity, scene dynamics, or shot boundaries vary over time. Adaptive strategies form variable-length segments that better allocate bits across the video (Ge et al., 2024; Yang et al., 2024). The most relevant work is M3-CVC (Wan et al., 2025): it segments video into variable-length clips based on semantic changes and optical flow, and reconstructs each clip from a decoded keyframe together with a text description using a diffusion-based generative model. While promising, text-based conditioning can be too coarse or ambiguous to capture fine-grained motion and complex dynamics, motivating the use of compact sparse trajectories as a more explicit conditioning signal for diffusion-based reconstruction.

Domain-specific ultra-low-bitrate video communication has also been explored using compact semantic control signals, such as facial landmarks or keypoints for conferencing (Oquab et al., 2021; Konuko et al., 2021) and parametric human-body representations (Chen et al., 2025). These approaches exploit strong structural priors about the content (e.g., faces or articulated bodies) to achieve extreme compression. However, because they rely on domain-specific assumptions, they are not directly comparable to methods targeting unconstrained natural videos, which is the setting we consider.

## 3 Proposed Approach: Concept Overview

In this section, we provide a conceptual overview of our proposed generative video compression algorithm. We couple *active sampling* of keyframes and intermediate context at the encoder with a *conditional diffusion* decoder. The sampled keyframes and selected context create a content-adaptive temporal segmentation of the video where the decoding is done in blocks of variable length. The decoder treats the selected context as side information and reconstructs the intermediate frames between each pair of consecutive keyframes

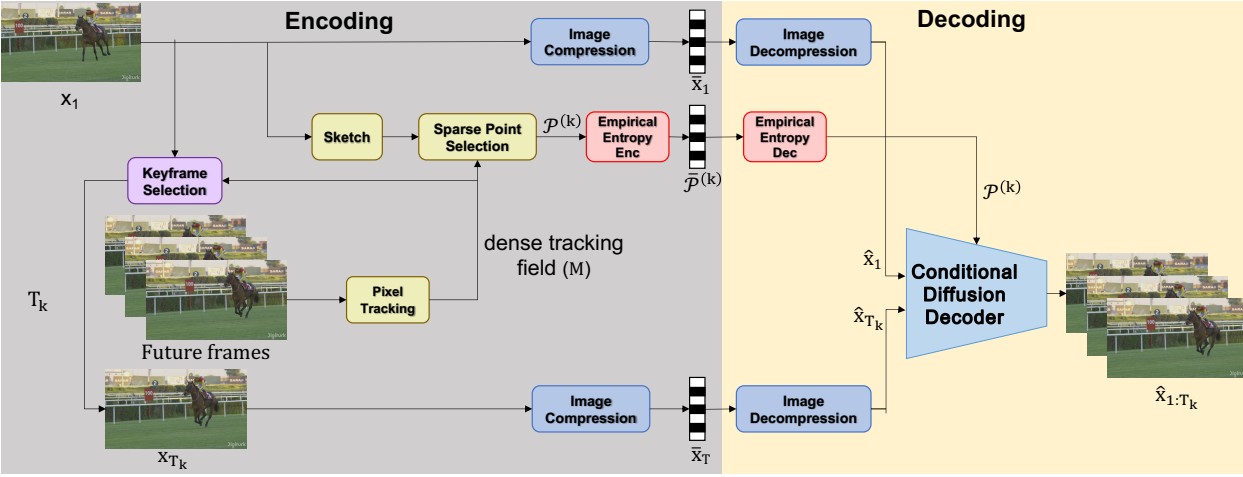

Figure 1: **Framework of ActDiff-VC.** Given the first frame, a dense point tracker estimates the dense tracking field **M** across subsequent frames. The sparse point selector, guided by a sketch of the first frame, subsamples the dense tracking field to form the conditioning sparse trajectory set $\mathcal{P}^{(k)}$. On the decoder side, the diffusion model is conditioned on $\mathcal{P}^{(k)}$ together with the first and last frames to reconstruct the video.

by running a reverse diffusion process conditioned on this received side information (in the form of sparse trajectory conditioning). Figure 1 illustrates our proposed pipeline.

### 3.1 Detailed Overview

We decompose each input video into a sequence of *content-adaptive* segments, also known as groups of pictures (GOPs), indexed by $k = 1, \ldots, K$. GOP segment $k$ contains $T_k$ frames $\{x_t^{(k)}\}_{t=1}^{T_k}$ with $x_t^{(k)} \in \mathbb{R}^{3 \times H \times W}$ and a one-frame overlap with the preceding and subsequent segments, i.e., $x_{T_k}^{(k)} = x_1^{(k+1)}$. At a high level, the encoder observes the video stream and chooses when to sample densely through keyframes and when to sample sparsely through compact motion conditioning signals. The decoder, on the other hand, relies on a conditional diffusion model to generate the missing information across each segment.

To guide both sampling decisions and the diffusion decoder, we use motion information from video tracking. Let $\Omega \subset \mathbb{Z}^2$ denote the set of pixel coordinates in the first frame $x_1^{(k)}$. The dense point tracker estimates, for each pixel $p \in \Omega$, a sequence of 2D displacements from the initial frame at each subsequent frame $t$:

$$\mathbf{M} : \Omega \to \mathbb{R}^{2T_k}, \quad p \mapsto \mathbf{M}(p) \equiv \left( \mathbf{u}_1(p), \ldots, \mathbf{u}_{T_k}(p) \right), \tag{1}$$

where $\mathbf{u}_t(p) \in \mathbb{R}^2$ denotes the displacement vector of pixel $p$ from its position in the initial frame $x_1^{(k)}$ to its position at frame $t$. This dense tracking field **M** captures the motion and temporal variation throughout the $k$-th segment, allowing the model to represent both local and global scene dynamics. While pixel tracking offers several advantages over traditional motion representations, its most important advantage for us is the high spatial correlation. More specifically, in natural videos, the trajectory of the pixel $p$, $p \mapsto \mathbf{M}(p)$, is highly correlated with the trajectory of its neighboring pixel $q$, $q \mapsto \mathbf{M}(q)$. This inherently high spatial correlation allows for further sub-sampling of the trajectory to a sparse pixel set $S \subset \Omega$ without negatively impacting motion fidelity while guaranteeing an ultra-low bit rate. We therefore introduce the *sparse trajectory set* for the $k$-th segment as the subsampling of the dense tracking field **M** onto $S$,

$$\mathcal{P}^{(k)} := \left\{ \left( q, \{\mathbf{u}_t(q)\}_{t=1}^{T_k} \right) : q \in S \right\}. \tag{2}$$

Our conditional diffusion model decoder operates on each segment independently, relying on the frame overlaps to effectively regulate the temporal variations across GOP segments. Given the decoded boundary keyframes $\hat{x}_1^{(k)}$, $\hat{x}_{T_k}^{(k)}$ and sparse trajectory set $\mathcal{P}^{(k)}$, , our decoder synthesizes a length-$T_k$ sequence of frames whose appearance matches $\hat{x}_1^{(k)}$, $\hat{x}_{T_k}^{(k)}$ and whose motion follows $\mathcal{P}^{(k)}$. For notational simplicity, we omit

segment superscripts on segment-dependent quantities whenever the segment index is clear from context. The final component handles compression. Keyframes are compressed with a lossy image compressor, whereas the sparse trajectory set is compressed losslessly with an entropy coder. The transmitted bitstream therefore comprises (i) compressed keyframes, (ii) losslessly compressed sparse trajectory set, and (iii) the segment size. At the receiver, the decoder first decompresses the keyframes and the sparse trajectory set; it then uses the keyframes together with $\mathcal{P}^{(k)}$ to guide diffusion-based synthesis of each segment. The following subsections describe each component conceptually, while Section 4 describes our concrete implementation.

## 3.2 Video Tracking as Sparse Trajectory Conditioning

To guide the diffusion decoder's motion synthesis, we represent temporal dynamics using dense pixel tracking. Rather than transmitting conventional motion vectors or optical flow fields, we leverage recent advances in dense point tracking (e.g., (Harley et al., 2025)) to estimate a dense tracking field $\mathbf{M}$ over each segment. These trajectories capture how each pixel in the reference keyframe $x_1^{(k)}$ moves through subsequent frames, providing a rich motion signal that the diffusion model can condition on during generation. Pixel tracking offers several advantages over traditional motion representations. First, tracking naturally handles long-range motion and maintains temporal consistency across multiple frames, whereas frame-to-frame motion vectors can accumulate drift (Neoral et al., 2024; Harley et al., 2025). Second, the dense tracking field provides explicit correspondence information that helps the diffusion model understand which regions of the reference keyframe $x_1^{(k)}$ should appear at which locations in future frames, reducing ambiguity in the generation process. Furthermore, tracking is inherently sparse-compatible. That is, given a sparse set $S$ of selected points in the initial frame, we estimate dense displacements via a radial basis function (RBF) kernel:

$$\widehat{\mathbf{M}}(p \mid S) = \sum_{q \in S} \alpha(p, q)\, \mathbf{M}(q), \qquad \alpha(p, q) = \frac{\exp\left(-\frac{\|p-q\|^2}{2\sigma^2}\right)}{\sum_{q' \in S} \exp\left(-\frac{\|p-q'\|^2}{2\sigma^2}\right)}. \tag{3}$$

where $\|p - q\|$ denotes the Euclidean distance between pixels $p$ and $q$, and $\sigma$ is the kernel bandwidth. We assume local spatial correlation in the tracking field: pixels that are close in the initial frame $x_1^{(k)}$ are more likely to have similar motion trajectories, especially when they lie in the same motion region. We can find $S \subset \Omega$ with $|S| \leq B$ such that the weighted reconstruction error:

$$\mathcal{R}(S) \equiv \sum_{p \in \Omega} w^{(k)}(p) \left\|\mathbf{M}(p) - \widehat{\mathbf{M}}(p \mid S)\right\|_2^2, \tag{4}$$

can be small, where $w^{(k)} : \Omega \to \mathbb{R}_{\geq 0}$ is an importance weight derived from the first frame (e.g., edge magnitude or gradient saliency).

## 3.3 Conditional Diffusion Decoder

Our compression framework relies on a conditional video diffusion model that serves as the decoder. Unlike traditional codecs that reconstruct frames through explicit transform coding of residuals, our approach leverages the generative capabilities of diffusion models to synthesize video content from minimal conditioning signals. Ideally, such a conditional diffusion model is trained from scratch; however, it can also be built by adapting existing diffusion-based video-generation models. Section 4.1 provides a detailed description of this model and how we adapt it for video compression at ultra-low bit rate.

Our proposed diffusion model takes as input: (i) conditioning image latents from the first and last frames of each GOP segment, providing appearance anchors, and (ii) the sparse trajectory set $\mathcal{P}^{(k)}$ that encodes motion dynamics throughout the temporal extent of the segment. The diffusion process iteratively refines an initial noisy latent representation toward a high-quality video sequence that matches both the appearance constraints (from keyframes) and the motion constraints (from tracking). By conditioning the diffusion process on both appearance and motion, the model can generate high-fidelity video sequences that respect the scene's motion structure while maintaining visual fidelity to the keyframe appearances. Furthermore, frame overlap ensures smooth transitions across consecutive GOP segments.

### 3.4 Content-Adaptive Keyframe Selection

A fundamental challenge in video compression is determining optimal keyframe placement. Traditional codecs often use fixed GOP structures with predetermined keyframe intervals, which fail to account for varying scene complexity, motion patterns, and visual content across different video segments.

Our approach introduces content-adaptive keyframe selection that determines GOP segment boundaries based on how long appearance and motion information from a keyframe remain informative for reconstruction. In particular, given the dense tracking field $\mathbf{M}$ from the first frame of the segment $x_1^{(k)}$, we forward-splat the first frame $x_1^{(k)}$ along the displacements $\{\mathbf{u}_t(p)\}_{p \in \Omega}$ to obtain warped images $\tilde{x}_t^{(k)}$. The overlap between the warped and original frames, together with their perceptual similarity, is used to compute a keyframe-selection score $\theta_t$, which serves as a surrogate for how informative the current keyframe remains for reconstructing future frames. The precise definition of $\theta_t$ is given in Section 4.

## 4 Proposed Approach: Implementation

This section describes the concrete implementation of the conceptual components introduced in Section 3. We begin by detailing the conditional video diffusion model and how it is adapted for compression. We then present our implementations of the content-adaptive keyframe selection, the corresponding budget-aware sparse trajectory selection, and encoding of all transmitted components.

### 4.1 Conditional Diffusion Decoder Implementation

Our conditional diffusion decoder is built upon DaS (Gu et al., 2025), a transformer-based video diffusion model that operates in a latent space. Given a conditioning image $x_1$ and a sparse trajectory set $\mathcal{P}$ of temporal length $T$, DaS synthesizes a length-$T$ video whose appearance matches $x_1$ and whose motion follows $\mathcal{P}$. The model uses temporal and spatial downsampling factors $s_T = 4$ and $s_S = 8$. For notational clarity, we define $T_{\text{lat}} := \left\lceil \frac{T}{s_T} \right\rceil$, $h := \left\lceil \frac{H}{s_S} \right\rceil$, $w := \left\lceil \frac{W}{s_S} \right\rceil$. A motion VAE encoder $\mathcal{E}_{\text{trk}}$ maps the sparse tracking set $\mathcal{P}$ to a latent tensor $z_{\text{trk}} = \mathcal{E}_{\text{trk}}(\mathcal{P}) \in \mathbb{R}^{T_{\text{lat}} \times h \times w \times 16}$. The construction of this motion VAE encoder is described in Appendix A8. Separately, the VAE encoder $\mathcal{E}$ maps the conditioning image $x_1$ to a single-frame latent $z_1 = \mathcal{E}(x_1) \in \mathbb{R}^{1 \times h \times w \times 16}$. The image latent is temporally expanded by placing it in the first latent time slice and zero-padding the remaining slices, yielding $z_{1,\text{pad}} \in \mathbb{R}^{T_{\text{lat}} \times h \times w \times 16}$. The denoising diffusion transformer (DiT) takes the concatenation of $z_{1,\text{pad}}$ with Gaussian noise and iteratively denoises it over multiple reverse diffusion steps. In parallel, a conditioning DiT processes the tracking latent $z_{\text{trk}}$ and extracts motion features at each transformer block. These motion features are linearly projected and injected into the corresponding layers of the denoising DiT via additive conditioning, allowing the diffusion process to be guided by the sparse motion signal. Finally, the VAE decoder $\mathcal{D}$ maps the denoised latent back to pixel space to produce the reconstructed frames.

**Bidirectional Conditioning.** To mitigate temporal discontinuities at segment boundaries, we use the pretrained DaS model with bidirectional boundary conditioning at inference time, without additional training or fine-tuning. Let $x_1$ and $x_{T_k}$ denote the segment's boundary frames, with corresponding latents $z_1$ and $z_{T_k}$. We form a *dual-conditioned latent stack*:

$$z_{\text{dual}} = \left[ z_1, \underbrace{\mathbf{0}, \ldots, \mathbf{0}}_{T_{\text{lat}}-2 \text{ interior zero slices}}, z_{T_k} \right], \tag{5}$$

where each zero slice $\mathbf{0} \in \mathbb{R}^{h \times w \times 16}$ has the same spatial and channel dimensions as a single latent frame. This dual-conditioned stack replaces the single-keyframe stack $z_{1,\text{pad}}$ used in the original DaS formulation. The denoising DiT then operates on the concatenation of $z_{\text{dual}}$ with Gaussian noise, encouraging the generated sequence to remain consistent with both boundary keyframes. As adjacent segments share a one-frame overlap, this bidirectional conditioning introduces minimal rate overhead while improving temporal smoothness.

**Adapting a fixed-length generator to variable-length segments.** Although DaS produces a fixed-length sequence, our compression system operates on variable-length segments of length $T_k$. To reconcile these, we temporally interpolate the sparse trajectory set for segment $k$ onto the model's fixed temporal grid. After generation, we subsample the generated sequence to $T_k$ frames.

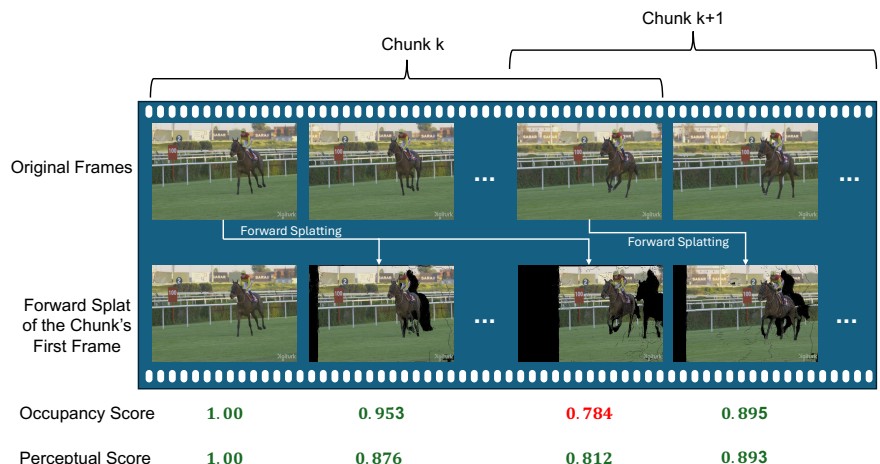

Figure 2: **Content-Adaptive Keyframe Selection.** The first frame of each segment is forward-splatted through the next frames using a dense tracker, yielding target-space occupancy $\mathrm{occ}(t)$ and perceptual similarity $\mathrm{sim}_{\mathrm{perc}}(t)$. The next keyframe is selected at the earliest $t$ such that $\mathrm{occ}(t) < \theta_{\mathrm{occ}}$ or $\mathrm{sim}_{\mathrm{perc}}(t) < \theta_{\mathrm{perc}}$ holds for $L$ consecutive frames. In this visualization, $L = 1$ and $\theta_{\mathrm{occ}} = \theta_{\mathrm{perc}} = 0.8$. The next segment begins with a one-frame overlap, and the procedure repeats.

## 4.2 Pixel Tracking Implementation

Among candidate motion cues, including frame-to-frame optical flow, scene flow, and long-range point tracking, we adopt long-range point tracking to mitigate error accumulation from chaining short-horizon flows and to obtain temporally coherent trajectories over each segment. Specifically, we employ AllTracker (Harley et al., 2025), which estimates the dense tracking field $\mathbf{M}$ from the segment's initial frame to all subsequent frames; for each pixel in the initial frame, it provides a trajectory specifying its corresponding location at each time step, together with per-frame visibility and confidence. AllTracker's reliance on its occlusion-aware, geometry-sensitive formulation yields trajectories that remain stable under large motions and viewpoint changes. This is precisely the behavior required by our diffusion decoder.

## 4.3 Content-Adaptive Keyframe Selection Implementation

In the encoder, we implement content-adaptive keyframe selection by computing occupancy and perceptual similarity metrics between the original video frames and the forward-splatted rendering of the keyframe for each potential segment endpoint. Given the dense tracking field $\mathbf{M}$ from the current keyframe, we forward-splat the keyframe $x_1^{(k)}$ along the displacements $\{\mathbf{u}_t(p)\}_{p \in \Omega}$ to obtain splatted images $\tilde{x}_t^{(k)}$ and binary occupancy masks:

$$O_t(q) = \mathbb{I}\Big[ \exists\, p \in \Omega : \; p + \mathbf{u}_t(p) = q \Big]. \qquad (6)$$

The occupancy fraction at time $t$ is

$$\mathrm{occ}(t) \;=\; \frac{1}{|\Omega|} \sum_{q \in \Omega} O_t(q), \qquad (7)$$

measuring what fraction of the target frame is covered by the forward-warped keyframe. For perceptual similarity, we compute

$$\mathrm{sim}_{\mathrm{perc}}(t) \;=\; 1 - \mathrm{LPIPS}\big(O_t \odot \tilde{x}_t^{(k)},\, O_t \odot x_t^{(k)}\big), \qquad (8)$$

where $\odot$ denotes element-wise multiplication and LPIPS (Zhang et al., 2018) is a learned perceptual similarity metric. Masking both images with $O_t$ restricts LPIPS to covered regions, avoiding penalties from uncovered areas. We then define the combined keyframe-validity score $\theta_t \;:=\; \min\left\{\frac{\mathrm{occ}(t)}{\theta_{\mathrm{occ}}}, \frac{\mathrm{sim}_{\mathrm{perc}}(t)}{\theta_{\mathrm{perc}}}\right\}$, where $\theta_{\mathrm{occ}}$ and $\theta_{\mathrm{perc}}$ are the occupancy and perceptual-similarity thresholds, respectively. We define the endpoint of segment $k$ as the earliest frame index $t > \tau_{k-1}$ such that this combined score remains below one for $L$ consecutive

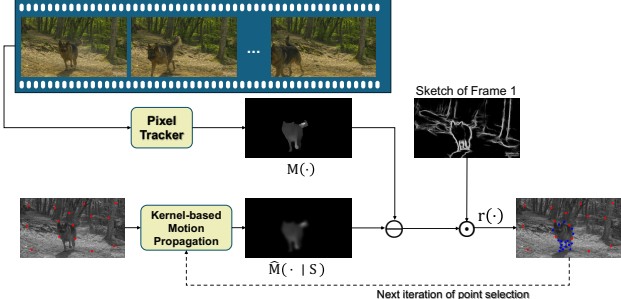

Figure 3: **Budget-Aware Sparse Trajectory Selection.** Given the current tracking set $S$ (red dots), the dense tracking field is estimated as $\widehat{\mathbf{M}}(\cdot \mid S)$ using the RBF kernel interpolation in equation 3. The residual $r(p)$, as defined in equation 11, quantifies the discrepancy between the dense tracking field $\mathbf{M}(\cdot)$ and its reconstruction $\widehat{\mathbf{M}}(\cdot \mid S)$. Points with the largest sketch-weighted residuals (blue dots) are added to $S$, improving motion reconstruction in underrepresented or high-error regions. This procedure repeats until the bit-rate budget is reached or the reconstruction error converges.

frames, i.e.,

$$\tau_k := \min_{t > \tau_{k-1}} \left\{ t \,\Big|\, \theta_{t+\ell} < 1 \quad \forall \ell \in \{0, \dots, L-1\} \right\}. \tag{9}$$

In other words, $\tau_k$ is the first frame in the earliest run of $L$ consecutive frames for which the keyframe no longer provides sufficient coverage or perceptual similarity. The parameter $L$ acts as a hysteresis term, preventing isolated score drops from triggering a new segment boundary. The corresponding segment length is $T_k = \tau_k - \tau_{k-1} + 1$. Figure 2 illustrates this process. Once a keyframe boundary is identified, the next segment begins with a one-frame overlap, and the procedure repeats. This implementation ensures that segment lengths adapt to scene content: static scenes yield longer segments with fewer keyframes, while dynamic scenes with occlusions or shot changes yield shorter segments with more frequent keyframes, thereby balancing keyframe coding cost against reconstruction quality.

### 4.4 Sparse Trajectory Conditioning Implementation

While the dense tracking field $\mathbf{M}$ provides complete motion information, transmitting all pixel trajectories is prohibitively expensive. We therefore introduce a budget-aware point selection strategy that identifies a sparse subset $S \subset \Omega$ of maximally informative tracks. We seek a subset $S \subset \Omega$ with $|S| \leq B$ that approximately minimizes the weighted reconstruction objective $\mathcal{R}(S)$ in equation 4. We adopt a budget-aware greedy selection that iteratively reduces $\mathcal{R}(S)$. To define the importance weights $w^{(k)}$, we apply Holistically-Nested Edge Detection (HED) (Xie & Tu, 2015) to the segment's first frame, yielding an edge map, which we refer to as the sketch of the frame:

$$w^{(k)}(p) = \text{HED}\left(x_1^{(k)}\right)(p), \quad p \in \Omega. \tag{10}$$

**Initialization.** We partition the image into a coarse grid and select the pixel with maximum sketch strength in each cell, ensuring initial spatial coverage of the tracking set.

**RBF Kernel Interpolation.** Given a current set $S$ of selected points, we estimate the dense displacement field $\widehat{\mathbf{M}}(\cdot \mid S)$ using the RBF kernel in equation 3. Specifically, the interpolation weights are determined by the Euclidean distance between pixels, so that nearby selected points have greater influence on the estimated motion field. Starting from the sketch-initialized grid $S_0$, we select the kernel bandwidth $\sigma$ by minimizing the weighted reconstruction objective $\mathcal{R}(S_0)$ in equation 4, and use this $\sigma$ for subsequent refinement of $S$.

**Greedy Refinement.** We refine $S$ iteratively by adding points at which the current reconstruction exhibits the largest residual. At iteration $m$, we compute a residual map

$$r(p) = \frac{1}{T_k} \sum_{t=1}^{T_k} w^{(k)}(p) \left\| \mathbf{M}_t(p) - \widehat{\mathbf{M}}_t(p \mid S_m) \right\|_2^2, \tag{11}$$

identify local maxima of $r(\cdot)$ over $\Omega$, and add the top candidates to form $S_{m+1}$. We then recompute $\widehat{\mathbf{M}}_t(\cdot \mid S_{m+1})$ and iterate. The process terminates when the budget is exhausted ($|S_m| = B$) or when $r(p)$ falls below a tolerance for all $p \in \Omega$. Figure 3 illustrates this iterative refinement. Upon termination of the iterative refinement, the resulting sparse trajectory set $\mathcal{P}^{(k)}$, as defined in equation 2, provides reliable tracking, strong spatial coverage, and high informativeness under the fixed budget constraint.

### 4.5  Entropy Coding of Components

We entropy-code the keyframes and sparse trajectory set to remove statistical redundancy. Keyframes $x_1^{(k)}$ and $x_{T_k}^{(k)}$ are compressed using HiFiC (Mentzer et al., 2020). For the sparse trajectory set $\mathcal{P}^{(k)}$, each selected point $q \in S$ is represented by its initial location $q$ together with temporal displacement differences $\Delta \mathbf{u}_t(q) = \mathbf{u}_t(q) - \mathbf{u}_{t-1}(q)$ for $t = 2, \ldots, T_k$. The sequence $\{\Delta \mathbf{u}_t(q)\}$ is then jointly entropy coded using a learned Huffman code derived from the empirical symbol distribution. The Huffman codebook is transmitted as part of the bitstream, and its overhead is included in the reported bitrate. Because motion deltas exhibit temporal correlation, this coding achieves significant compression. The initial points $\{q : q \in S\}$ and the segment length $T_k$ are transmitted as side information without additional entropy coding.

## 5  Evaluation

### 5.1  Experimental Setup and Evaluation Metrics

We evaluate our approach on the UVG (Mercat et al., 2020) and MCL-JCV (Wang et al., 2016) datasets. We report perceptual quality using LPIPS (Zhang et al., 2018), FID (Heusel et al., 2017), KID (Bińkowski et al., 2018), and NIQE (Mittal et al., 2012), which are standard metrics for evaluating generative reconstruction. We focus our main evaluation on perceptual metrics, as our goal is to assess visual realism in the ultra-low-bitrate regime, where pixel-wise fidelity is less informative. For completeness, distortion-oriented metrics and full-reference temporal consistency metrics are reported in Appendices A9 and A10, respectively.

**Compared Methods.** We compare our method against several learned video compression baselines, including DCVC (Li et al., 2021), DCVC-TCM (Sheng et al., 2022), DCVC-HEM (Li et al., 2022), DCVC-DC (Li et al., 2023), and DCVC-FM (Li et al., 2024b), as well as the GAN-based method PLVC (Yang et al., 2022).[1] We evaluate these methods using their official codebases and pretrained weights on both datasets. ActDiff-VC and all methods included in the main comparison are evaluated at a resolution of $480 \times 720$. For completeness, we provide a separate comparison with the diffusion-based method EVC-PDM (Li et al., 2024a) in Appendix A5. Because the released EVC-PDM implementation supports only $128 \times 128$ inputs, we report its results separately rather than including them in the resolution-matched main comparison.

### 5.2  Experimental Results

**Quantitative Comparison.** Figure 4 compares ActDiff-VC with existing methods on the UVG and MCL-JCV datasets in terms of LPIPS, FID, KID, and NIQE (lower is better for all metrics). We focus on perceptual metrics, as distortion metrics such as PSNR are less informative in the ultra-low-bitrate setting.

ActDiff-VC shows its clearest advantage in NIQE and KID in the ultra-low-bitrate regime. In terms of NIQE, ActDiff-VC reaches 5.08 at 0.0164 BPP on UVG and 5.10 at 0.0276 BPP on MCL-JCV, whereas DCVC-FM requires approximately 0.0372 BPP and 0.0780 BPP, respectively, to achieve comparable NIQE values. This corresponds to bitrate reductions of 55.9% on UVG and 64.6% on MCL-JCV. A similar trend is observed for KID: at nearly identical low bit rates, ActDiff-VC achieves 0.0168 at 0.0164 BPP on UVG and 0.0236 at 0.0276 BPP on MCL-JCV, improving over DCVC-FM by 64.6% and 58.0%, respectively.

FID also shows consistent gains in the ultra-low-rate regime. On UVG, ActDiff-VC obtains an FID of 31.94 at 0.0164 BPP, improving over DCVC-FM by 37.7% at a similar bitrate, while on MCL-JCV it achieves an FID of 53.40 at 0.0276 BPP, improving by 22.8% at a comparable rate. These results indicate that ActDiff-

---

[1]We select these methods because their official codebases and pretrained weights are publicly available, enabling reproducible evaluation.

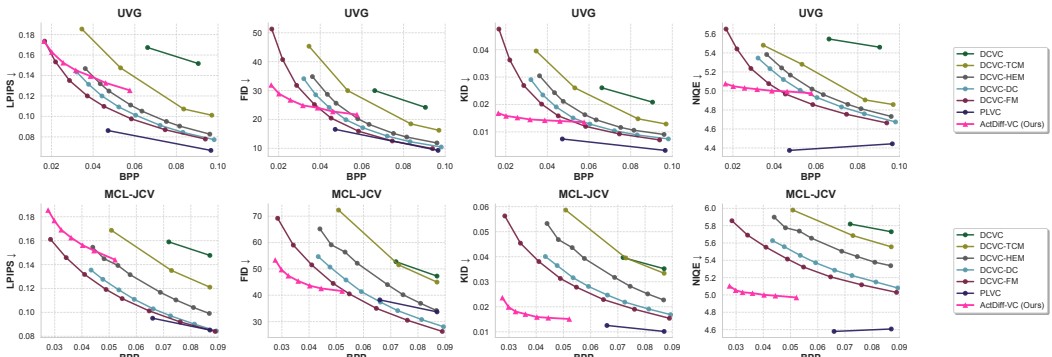

Figure 4: Quantitative comparison on the UVG and MCL-JCV datasets. We report LPIPS, FID, KID, and NIQE as a function of bits per pixel (BPP); lower values indicate better performance for all metrics.

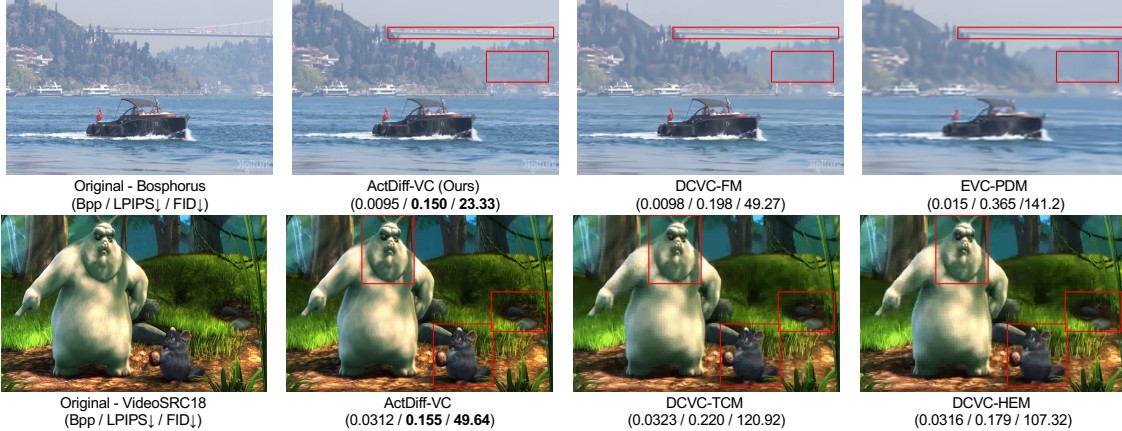

Figure 5: Qualitative comparison on representative sequences from UVG and MCL-JCV. We compare ActDiff-VC against competing methods at comparable, and in some cases higher, bit rates. The numbers below each reconstruction report per-video BPP, LPIPS, and FID (lower is better). Red boxes highlight challenging regions where competing methods exhibit stronger blur, artifacts, or structural degradation, while ActDiff-VC better preserves fine details, textures, and boundaries.

VC consistently improves distributional and no-reference perceptual metrics such as FID, KID, and NIQE at extremely low bitrates. For LPIPS, ActDiff-VC is not the best overall but remains competitive in the low-rate regime. On UVG, at 0.0460 BPP, ActDiff-VC achieves an LPIPS of 0.1328, outperforming DCVC (0.1674 at 0.0662 BPP) and DCVC-TCM (0.1475 at 0.0532 BPP), while remaining higher than DCVC-FM (0.0976 at 0.0584 BPP). A similar pattern is observed on MCL-JCV, where ActDiff-VC achieves an LPIPS of 0.1516 at 0.0444 BPP, outperforming DCVC (0.1589 at 0.0719 BPP), although it remains behind DCVC-FM. Compared with the GAN-based PLVC, which operates primarily at higher bit rates, ActDiff-VC extends perceptual compression to a substantially lower bitrate regime while maintaining similar trends in FID, KID, and NIQE. We additionally report distortion-oriented metrics and full-reference temporal metrics, including PSNR, MS-SSIM, T-LPIPS, and PVCS, in Appendices A9 and A10, respectively.

**Qualitative Results.** Figure 5 shows qualitative comparisons on representative sequences from UVG and MCL-JCV. Across the selected examples, ActDiff-VC produces reconstructions with sharper local structure and fewer perceptual artifacts than competing methods, particularly in regions containing fine textures, boundaries, and small details. These examples are consistent with the quantitative results, showing that our method maintains perceptually realistic reconstruction even at very low bit rates. Additional qualitative examples are provided in Figure A7.

**Keyframe Selection Analysis.** Figure 6 illustrates the effect of our content-adaptive keyframe selection on videoSRC20 from MCL-JCV, where a scene cut occurs between Frame 42 and Frame 43. ActDiff-VC correctly identifies this transition and inserts a new keyframe, yielding a faithful reconstruction of the new scene immediately after the cut. In contrast, PLVC uses a fixed keyframe interval and cannot adapt to the

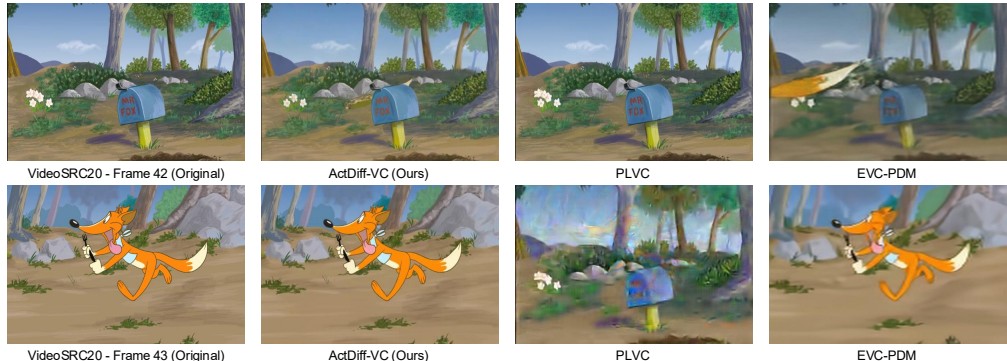

Figure 6: Visualization of content-adaptive keyframe selection. We show a frame before a scene change and the first frame after the cut, together with reconstructions from ActDiff-VC, PLVC, and EVC-PDM. ActDiff-VC detects the transition and inserts a new keyframe, producing a high-quality reconstruction after the cut. In contrast, PLVC relies on a fixed keyframe interval and fails to adapt to the scene cut, so the reconstructed frame after the cut still resembles the previous scene and contains visible artifacts. EVC-PDM can also adapt to scene changes via diffusion-based forecasting at the encoder, but still produces lower visual quality than ActDiff-VC.

| Components | | | Metrics (vs. baseline) | |
|---|---|:---:|:---:|:---:|
| Keyframe policy | Point selection strategy | Bidirectional cond. | $\Delta$LPIPS $\downarrow$ | $\Delta$FID $\downarrow$ |
| *Adaptive GOP (baseline block)* | | | | |
| Adaptive | Content-aware (sketch-weighted) | ✓ | 0.0000 | 0.0000 |
| Adaptive | Content-aware (sketch-weighted) | ✗ | +0.0321 | **+1.3356** |
| Adaptive | Uniform grid | ✓ | +0.0246 | +5.6126 |
| Adaptive | High-mag flow | ✓ | +0.0458 | +12.4746 |
| Adaptive | Content-aware (no sketch) | ✓ | **+0.0219** | +1.4193 |
| *Fixed GOP (comparison block)* | | | | |
| Fixed | Content-aware (no sketch) | ✓ | **+0.1323** | **+24.7450** |
| Fixed | Uniform grid | ✓ | +0.1336 | +37.9850 |
| Fixed | High-mag flow | ✓ | +0.1412 | +41.2817 |
| Fixed | Uniform grid | ✗ | +0.1764 | +59.4039 |

Table 1: Ablations at fixed bitrate. $\Delta > 0$ indicates degradation vs. the baseline (Adaptive GOP + Content-aware (sketch-weighted) + bidirectional conditioning); lower is better. Best non-baseline ablation values within each GOP block are **bold**.

abrupt content change, causing the reconstructed frame after the cut to resemble the previous scene with visible artifacts and distortions. EVC-PDM can also respond to scene changes, but does so through diffusion-based forecasting at the encoder while still producing lower visual quality than ActDiff-VC. This example shows that content-adaptive keyframe selection is effective for handling abrupt temporal discontinuities while preserving reconstruction quality. Additional examples are provided in Figure A8 and Figure A9.

## 5.3 Ablation Study

**Component Ablations.** Table 1 reports ablations of three key components on UVG at a fixed bitrate: (i) keyframe policy (Fixed GOP vs. Adaptive GOP), (ii) trajectory selection strategy, and (iii) bidirectional conditioning. We report metric deltas relative to a baseline configuration (Adaptive GOP + content-aware (sketch-weighted) trajectory selection + bidirectional conditioning), where $\Delta > 0$ indicates degradation and lower is better. For the Fixed GOP setting, the keyframe interval is set to the mean segment length produced by Adaptive GOP. Overall, Adaptive GOP and content-aware trajectory selection provide the largest gains, while bidirectional conditioning further improves reconstruction quality.

*Effect of bidirectional conditioning.* Bidirectional conditioning consistently improves visual quality across all settings. Under a Fixed GOP with uniform sampling, adding bidirectional conditioning reduces $\Delta$FID from +59.4039 to +37.9850 and $\Delta$LPIPS from +0.1764 to +0.1336, indicating its importance for stabilizing generative reconstruction.

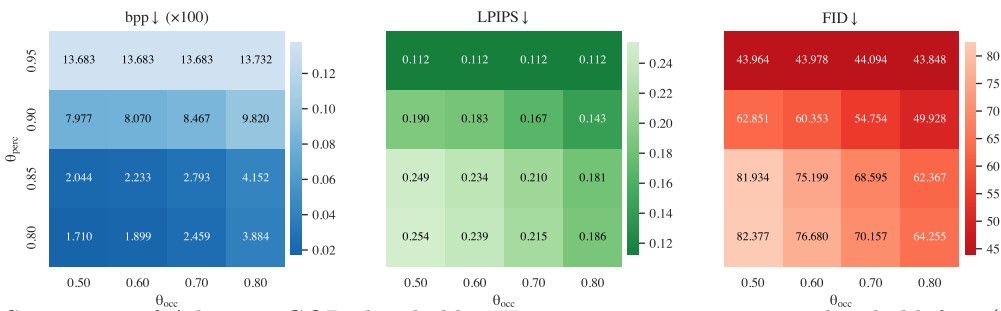

Figure 7: Sensitivity of Adaptive GOP thresholds. Heatmaps over occupancy threshold $\theta_{occ}$ (x-axis) and perceptual threshold $\theta_{perc}$ (y-axis), showing bitrate (bpp, ×100), LPIPS, and FID (lower is better). Increasing either threshold improves perceptual quality at the cost of higher bitrate.

*Effect of keyframe policy.* Replacing a Fixed GOP with Adaptive GOP substantially reduces degradation in both LPIPS and FID under matched settings. For example, under a uniform grid with bidirectional conditioning, $\Delta$FID falls from +37.9850 (Fixed GOP) to +5.6126 (Adaptive GOP), while $\Delta$LPIPS decreases from +0.1336 to +0.0246.

*Effect of trajectory selection.* Trajectory selection plays a key role in determining reconstruction quality. High-magnitude flow sampling over-focuses on large displacements while undersampling low-motion regions that are important for preserving texture, leading to worse LPIPS and FID compared to uniform sampling (e.g., $\Delta$FID +12.4746 vs. +5.6126 under Adaptive GOP with bidirectional conditioning). Uniform sampling is more balanced but content-agnostic, as it does not account for motion reconstruction difficulty. In contrast, the proposed content-aware (sketch-weighted) selection achieves the best performance by prioritizing informative regions. Removing sketch weighting leads to small but consistent degradation ($\Delta$FID +1.4193, $\Delta$LPIPS +0.0219), showing that incorporating structural priors improves robustness. Without this weighting, displacement noise in low-texture regions propagates more easily and introduces visible artifacts.

**Hyperparameter sensitivity for Adaptive GOP.** Figure 7 shows the sensitivity of the Adaptive GOP procedure to its two termination thresholds, $\theta_{occ}$ and $\theta_{perc}$, on high-motion MCL-JCV sequences with scene cuts. We evaluate $\theta_{perc}$ in a high-value range, as lower values rarely trigger keyframe insertion and therefore have limited impact on GOP segmentation. Both thresholds control the trade-off between bitrate and perceptual quality: increasing either threshold results in more conservative segment termination, yielding shorter GOPs (more frequent keyframes), higher bitrate, and improved reconstruction quality. Overall, $\theta_{occ}$ has a stronger and more consistent effect on the rate–quality trade-off. Increasing $\theta_{occ}$ leads to steady increases in bitrate and corresponding improvements in LPIPS and FID across all values of $\theta_{perc}$. In contrast, $\theta_{perc}$ has a more pronounced effect only in the high-threshold regime, where increasing it from 0.85 to 0.90 or higher causes a sharp increase in bitrate together with significant gains in perceptual quality. Given our target operating regime of ultra-low bitrate compression (bpp $\leq 0.05$), we select $\theta_{perc} = 0.85$ and $\theta_{occ} = 0.8$ in all experiments. This setting achieves 0.0415 bpp while maintaining strong perceptual quality, providing a favorable trade-off between compression efficiency and reconstruction fidelity.

# 6 Conclusion

In this work, we introduced **ActDiff-VC**, a diffusion-based video compression framework designed for the ultra-low-bitrate regime. Our approach combines a pre-trained conditional diffusion decoder with an active sampling strategy that selects sparse and informative conditioning signals for reconstruction. This design enables strong perceptual reconstruction quality at extreme compression rates. Experiments on standard benchmarks demonstrate the effectiveness of **ActDiff-VC** in this challenging setting, and ablation studies confirm the importance of its main components. Overall, our results suggest that diffusion-based generative models offer a promising direction for perceptual video compression in the ultra-low-bitrate regime.

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

## A1   Appendix Overview

This appendix provides additional discussion and analysis to complement the main paper. Section A2 discusses the main limitations of our method and directions for improving efficiency and robustness. Section A3 analyzes the encoding and decoding latency of the proposed method in comparison with competing codecs, highlighting the favorable encoder runtime of ActDiff-VC and the additional decoding cost introduced by diffusion-based reconstruction. Section A4 studies the quality–efficiency trade-off associated with the number of reverse diffusion steps and motivates the 20-step configuration used in our experiments. Section A5 provides a separate comparison with EVC-PDM at the common evaluation resolution of $128 \times 128$, since its released implementation does not support the $480 \times 720$ resolution used in the main comparison. Section A6 studies the effect of the sparse trajectory budget on compression performance, showing how the number of transmitted trajectories influences bitrate and perceptual quality. Section A7 evaluates robustness to tracking inaccuracies through temporal subsampling, motion blur, and persistent drift in the sparse trajectory conditioning. Section A8 describes the motion encoder construction used in our implementation: the transmitted sparse trajectory set is deterministically converted into a synthetic video representation, which is then encoded into the latent motion-conditioning input used by the diffusion decoder. Section A9 reports PSNR and MS-SSIM as complementary distortion-oriented measures of reconstruction fidelity. Section A10 evaluates reference-aligned temporal fidelity using T-LPIPS and PVCS and discusses their interpretation for generative video reconstruction. Section A11 presents absolute visual-quality ratings and pairwise preference results from our human study, providing a complementary subjective evaluation of perceptual quality and temporal consistency. Finally, Section A12 provides additional qualitative comparisons that extend the visual results in the main paper, together with further examples of the proposed content-adaptive keyframe selection under abrupt scene transitions.

## A2   Limitations and Future Work

A primary limitation of ActDiff-VC is the computational cost of diffusion-based decoding. Compared with conventional feed-forward learned codecs, iterative denoising introduces substantially higher decoding latency and currently limits real-time deployment. Our diffusion-step analysis in Appendix A4 shows that reducing the number of sampling steps can substantially decrease decoding time while preserving competitive perceptual quality. Nevertheless, even with accelerated sampling, the decoder remains slower than conventional learned codecs. This computational trade-off is most relevant in the ultra-low-bitrate regime, where distortion-optimized codecs frequently produce overly smooth or blurry reconstructions, and additional decoder computation can instead be used to recover perceptually realistic content from highly compact conditioning signals.

A second limitation is weaker alignment with the exact temporal evolution of the source video. As shown by the full-reference temporal metrics in Appendix A10, ActDiff-VC obtains higher T-LPIPS and PVCS values than predictive learned codecs. This is expected in part because intermediate frames are generated from compressed boundary keyframes and a sparse trajectory set rather than reconstructed from explicitly coded per-frame motion and residual information. Consequently, local motion, texture evolution, or appearance changes may differ from the corresponding source frames. These results indicate reduced reference-aligned temporal fidelity, although they do not necessarily imply an equivalent reduction in perceived temporal coherence. Indeed, the human evaluation in Appendix A11, in which participants were asked to consider temporal consistency and motion quality in addition to overall visual quality, shows a clear preference for ActDiff-VC over the evaluated baselines.

Future work should therefore improve both decoding efficiency and temporal fidelity. Accelerated diffusion techniques, including model distillation, consistency models, and more efficient sampling schedules, provide promising directions for reducing inference time. Stronger temporal conditioning, explicit cross-segment consistency mechanisms, and joint optimization of tracking, trajectory compression, and generative reconstruction may further improve alignment with the source dynamics while increasing robustness to tracking errors.

The current computational profile of ActDiff-VC is particularly well suited to applications in which videos are encoded once using a lightweight encoder and reconstructed with access to greater decoder-side computation. Relevant settings include cloud-assisted video reconstruction, archival storage, offline content distribution, and one-to-many delivery systems. In these scenarios, the very low transmission rate and improved perceptual quality can justify allocating additional computation to the decoder. Extending these advantages to interactive and real-time applications through accelerated diffusion decoding remains an important direction for future work.

## A3    Encoding and Decoding Time Analysis

| Time per frame (ms) | DCVC | DCVC-TCM | DCVC-HEM | DCVC-DC | DCVC-FM | PLVC | ActDiff-VC (Ours) |
|---|---|---|---|---|---|---|---|
| Encoding ↓ | 1244 | 353 | 360 | 347 | 140 | 40539 | 109 |
| Decoding ↓ | 7137 | 129 | 81 | 126 | 116 | 141659 | 2311 |

Table A1: **Comparison of encoding and decoding time across competing methods.** We report the average encoding and decoding time per frame for all compared methods. For encoding time, we measure the runtime required to compress the input using each method and write the resulting bitstream to disk. For decoding time, we measure the runtime required to read the saved bitstream and reconstruct the frame. Lower is better for both metrics. ActDiff-VC achieves the fastest encoding time among all compared methods in this setting. Its decoding time is higher than conventional learned codecs, reflecting the cost of diffusion-based generation, but it remains substantially faster than the GAN-based PLVC.

Table A1 compares the average per-frame encoding and decoding latency of ActDiff-VC against competing methods.[2] All experiments were conducted on a single NVIDIA A100 GPU. For ActDiff-VC, encoding time includes dense point tracking, content-adaptive keyframe selection, HED-based sketch extraction, budget-aware sparse trajectory selection, keyframe compression, and entropy coding of the transmitted side information. To measure encoding time, we run each codec to compress the input frames and save the produced bitstream. To measure decoding time, we read the saved bitstream and time the reconstruction process at the decoder. The results show that ActDiff-VC has the fastest encoder among all compared methods, requiring only 109 ms per frame. This suggests that our compression pipeline is efficient on the encoder side despite relying on content-adaptive segmentation and sparse trajectory extraction.

On the decoder side, ActDiff-VC requires 2311 ms per frame. Although this is slower than the more efficient feed-forward learned codecs, it is faster than the original DCVC and substantially faster than the GAN-based PLVC. The higher decoding latency is expected because ActDiff-VC reconstructs frames through an iterative conditional diffusion process rather than direct feed-forward decoding. Overall, these results show that ActDiff-VC provides a highly efficient encoder, while concentrating most of its computational cost at the decoder. This trade-off motivates the use of ActDiff-VC in ultra-low-bitrate applications where achieving perceptually realistic video from severely limited transmitted information is more important than minimizing decoding time.

## A4    Diffusion Step Trade-Off

The iterative reverse diffusion process is the main source of decoding latency in ActDiff-VC. The pretrained diffusion backbone uses 50 sampling steps as its default inference setting. We therefore evaluate the decoder using 50, 40, 30, 20, and 10 diffusion steps to study the trade-off between reconstruction quality and inference efficiency. For a controlled comparison, all configurations use the same encoded bitstreams, keyframe locations, sparse trajectory conditioning, sampler, and inference parameters; therefore, changing the number of steps does not affect the bitrate or encoder-side computation. We report results at fixed operating points of 0.0186 BPP on UVG and 0.038491 BPP on MCL-JCV.

---

[2]We do not include EVC-PDM in Table A1 because its released implementation only supports compression at a resolution of $128 \times 128$, whereas all other methods are evaluated at $480 \times 720$. Since this operating resolution is substantially smaller, its runtime is not directly comparable. For reference, EVC-PDM reports an average encoding time of 2308 ms per frame and decoding time of 1353 ms per frame.

| Diffusion Steps | UVG (0.0186 BPP) | | | | MCL-JCV (0.038491 BPP) | | | | Decoding Time (ms/frame) ↓ |
|---|---|---|---|---|---|---|---|---|---|
| | LPIPS ↓ | FID ↓ | KID ↓ | NIQE ↓ | LPIPS ↓ | FID ↓ | KID ↓ | NIQE ↓ | |
| 50 | 0.1708 | **29.51** | **0.0144** | 5.423 | 0.1782 | 50.25 | **0.0198** | **5.144** | 5528 |
| 40 | 0.1716 | 29.88 | **0.0144** | **5.422** | 0.1779 | 50.29 | 0.0203 | 5.150 | 4523 |
| 30 | 0.1717 | 30.20 | 0.0158 | 5.437 | 0.1746 | 49.08 | 0.0199 | 5.173 | 3356 |
| **20** | **0.1694** | 30.25 | 0.0161 | 5.442 | **0.1704** | **48.54** | 0.0207 | 5.151 | 2311 |
| 10 | 0.1844 | 36.19 | 0.0192 | 5.511 | 0.1764 | 52.63 | 0.0241 | 5.162 | 1205 |

Table A2: **Quality–efficiency trade-off for different numbers of diffusion steps.** Results are reported at fixed operating points of 0.0186 BPP on UVG and 0.038491 BPP on MCL-JCV. Reducing the sampling process from 50 to 20 steps lowers decoding time from 5528 to 2311 ms/frame while preserving strong perceptual quality across both datasets. The 20-step configuration achieves the best LPIPS on UVG and MCL-JCV and the best FID on MCL-JCV, while remaining close to the best KID and NIQE results. We therefore adopt 20 diffusion steps as the default configuration. Lower is better for all metrics.

Table A2 shows that decoding time decreases approximately with the number of diffusion steps, while the quality differences between 20 and 50 steps remain relatively small. In particular, using 20 steps reduces the decoding time by 58.2% compared with the original 50-step setting.

Importantly, this speed improvement does not result in a consistent reduction in reconstruction quality. On UVG, the 20-step configuration achieves the best LPIPS of 0.1694, while its FID, KID, and NIQE remain close to those obtained with longer sampling schedules. On MCL-JCV, it achieves both the best LPIPS and the best FID, with values of 0.1704 and 48.54, respectively. Its KID and NIQE also remain close to the best values produced by the 50-step setting.

By contrast, reducing the sampling process to 10 steps is too aggressive. Although it lowers the decoding time to 1205 ms/frame, it causes a noticeable increase in FID and KID on both datasets and also degrades LPIPS on UVG. Overall, 20 diffusion steps provide the most favorable balance between perceptual reconstruction quality and practical decoding time. We therefore use 20 steps as the default inference setting in ActDiff-VC.

## A5 Separate Comparison with EVC-PDM

The released implementation of EVC-PDM (Li et al., 2024a) supports video compression only at a spatial resolution of $128 \times 128$, whereas ActDiff-VC operates at $480 \times 720$, as dictated by its model architecture. We therefore exclude EVC-PDM from the resolution-matched comparison in the main paper and provide a separate evaluation here.

To establish a common evaluation resolution, we resize the videos reconstructed by ActDiff-VC and their corresponding source videos to $128 \times 128$ before computing the metrics. EVC-PDM is evaluated using its native $128 \times 128$ reconstructions and the same $128 \times 128$ source videos. The same resizing procedure is used for all frames and operating points. We report LPIPS, FID, KID, and PSNR, thereby covering both perceptual and distortion-oriented reconstruction quality. We do not report NIQE or MS-SSIM in this comparison because their standard implementations require greater spatial support and do not provide reliable estimates at a resolution of $128 \times 128$.

Figure A1 shows that ActDiff-VC consistently outperforms EVC-PDM across all four reported metrics on both datasets. On UVG, at nearby operating points of 0.0317 BPP for ActDiff-VC and 0.0332 BPP for EVC-PDM, ActDiff-VC reduces LPIPS by 52.4%, FID by 84.7%, and KID by 91.4%, while improving PSNR by 8.7 dB. A similar trend is observed on MCL-JCV: at 0.0360 BPP, compared with EVC-PDM at 0.0383 BPP, ActDiff-VC reduces LPIPS by 44.8%, FID by 75.0%, and KID by 90.6%, while improving PSNR by 8.1 dB. Thus, even when the reconstructions are assessed at the common $128 \times 128$ resolution, ActDiff-VC provides substantially better perceptual, distributional, and pixel-level reconstruction quality at comparable bitrates.

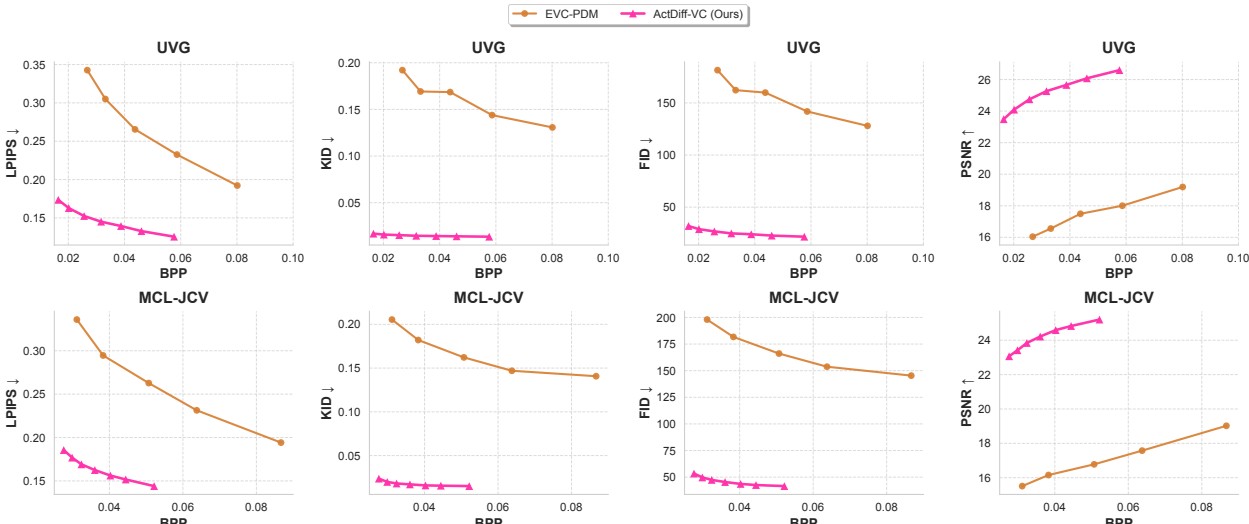

Figure A1: **Separate comparison with EVC-PDM on UVG and MCL-JCV at a common evaluation resolution of** $128 \times 128$. We report LPIPS, KID, FID, and PSNR as functions of bits per pixel (BPP). Lower values are better for LPIPS, KID, and FID, while higher values are better for PSNR. ActDiff-VC reconstructions and their corresponding source videos are resized to $128 \times 128$ before metric computation, while EVC-PDM is evaluated at the native resolution supported by its released implementation. ActDiff-VC consistently outperforms EVC-PDM across all reported metrics and operating points.

## A6  Impact of the Sparse Tracking Budget

| Point budget | bpp ↓ | LPIPS ↓ | FID ↓ |
|---|---|---|---|
| 40 | 0.017949 | 0.1666 | 37.780 |
| 80 | 0.018640 | 0.1598 | 36.939 |
| 120 | 0.019312 | 0.1523 | 35.288 |
| 160 | 0.019977 | 0.1481 | 34.616 |
| 200 | 0.020633 | 0.1463 | 34.174 |
| 250 | 0.021441 | 0.1463 | 33.720 |
| 300 | 0.022233 | 0.1460 | 33.409 |
| 350 | 0.023020 | 0.1455 | 33.275 |
| 500 | 0.025023 | 0.1451 | 32.596 |

Table A3: **Effect of the point budget on the compression–quality trade-off.** We vary the maximum number of transmitted point trajectories per segment while keeping all other components fixed. We report bitrate (bpp) together with perceptual quality metrics (LPIPS and FID) aggregated over the evaluation set. Increasing the point budget mildly increases bitrate while generally improving perceptual quality, with diminishing returns beyond moderate budgets. Lower is better for bpp, LPIPS, and FID.

Our method represents motion side information using a sparse trajectory set $\mathcal{P}^{(k)}$ ( equation 2), obtained by subsampling the dense tracking field $\mathbf{M}$. A key design parameter is the point budget $B$, which limits the maximum number of transmitted trajectories per GOP segment. To study its effect on compression performance, we sweep $B$ on the UVG dataset while keeping all other components fixed.

Table A3 shows the resulting bitrate together with LPIPS and FID. As the point budget increases, bitrate rises gradually because the majority of the total bitrate is consumed by keyframe encoding, so increasing the number of transmitted trajectories contributes only a relatively small additional cost. At the same time, reconstruction quality generally improves, with the largest gains appearing at smaller budgets and progres-

| Subsampling factor | LPIPS ↓ | FID ↓ | KID ↓ | NIQE ↓ |
|---|---|---|---|---|
| 1× | 0.1708 | 29.51 | 0.0144 | 5.423 |
| 2× | 0.1811 | 31.03 | 0.0153 | 5.541 |
| 4× | 0.1881 | 33.45 | 0.0161 | 5.658 |

Table A4: **Analysis of tracking robustness under temporal subsampling.** Increasing the subsampling factor exposes AllTracker to larger apparent inter-frame motion, yet ActDiff-VC exhibits only moderate and gradual degradation across all metrics, demonstrating robustness even at 4× subsampling. Lower is better for all metrics.

sively diminishing returns as $B$ increases. This behavior is consistent with the strong spatial correlation of dense trajectories discussed in Section 3, which allows a relatively small subset of tracked points to provide informative motion conditioning for reconstruction. Based on this trade-off, we use a point budget of $B = 300$ in the main experiments.

## A7 Robustness to Tracking Failures

ActDiff-VC uses a dense tracking field $\mathbf{M}$ to determine content-adaptive segment boundaries and select the sparse trajectory set $\mathcal{P}^{(k)}$ used to condition the diffusion decoder. We therefore study how inaccuracies in this motion representation propagate through the compression pipeline and affect reconstruction quality.

We consider three controlled perturbations: temporal subsampling of the video provided to AllTracker, motion blur applied only to the tracker input, and temporally correlated drift added directly to the selected sparse trajectories. The first two experiments evaluate failures originating during dense tracking and therefore allow the perturbation to affect both content-adaptive keyframe selection and budget-aware sparse trajectory selection. The third experiment keeps these upstream decisions unchanged and isolates the sensitivity of the conditional diffusion decoder to inaccurate sparse trajectory conditioning.

All remaining components and inference settings are kept fixed. The original, unperturbed videos are used for keyframe compression, reconstruction targets, and metric evaluation. Results are averaged over the seven UVG sequences using LPIPS, FID, KID, and NIQE, where lower values indicate better performance.

### A7.1 Temporal Subsampling of the Tracker Input

The goal of this experiment is to evaluate tracking robustness when consecutive observations contain larger apparent motion. For a temporal subsampling factor $s \in \{1, 2, 4\}$, AllTracker receives

$$\mathcal{X}^{(s)} = \{x_1, x_{1+s}, x_{1+2s}, \ldots\}, \tag{A1}$$

where $s = 1$ is the unmodified baseline. Increasing $s$ reduces the effective frame rate available to the tracker and increases the displacement between successive tracker observations, providing a controlled approximation of more challenging fast-motion conditions.

AllTracker estimates the dense tracking field from the temporally subsampled sequence. The resulting trajectories are then temporally interpolated to the original sequence length before being used for content-adaptive keyframe selection and budget-aware sparse trajectory selection. The resulting sparse trajectory set is subsequently compressed and passed to the conditional diffusion decoder, which reconstructs the original full-frame-rate video.

As shown in Table A4, all metrics degrade gradually as the temporal sampling rate available to AllTracker is reduced. Relative to the 1× baseline, 2× subsampling increases LPIPS by only 6.0%, FID by 5.2%, KID by 6.3%, and NIQE by 2.2%. Even under the more challenging 4× setting, where the tracker observes approximately four times larger inter-frame displacements, the corresponding increases remain limited to 10.1%, 13.4%, 11.8%, and 4.3%, respectively.

| Blur kernel size | LPIPS ↓ | FID ↓ | KID ↓ | NIQE ↓ |
|---|---|---|---|---|
| 1 | 0.1708 | 29.51 | 0.0144 | 5.423 |
| 11 | 0.1720 | 29.85 | 0.0148 | 5.483 |
| 21 | 0.1755 | 29.98 | 0.0153 | 5.532 |

Table A5: **Analysis of tracking robustness under motion blur.** Increasing the blur kernel size degrades the visual evidence available to AllTracker, yet ActDiff-VC exhibits only minor and gradual changes across all metrics, demonstrating strong robustness even under the larger kernel size of 21. Lower is better for all metrics.

These results demonstrate that ActDiff-VC is robust to substantially faster apparent motion in the tracker input. Even when AllTracker operates on a sequence subsampled by factors of 2× and 4×, the degradation remains limited across all perceptual metrics. In particular, LPIPS and NIQE change only modestly, indicating that the reconstructed videos preserve both perceptual similarity and visual naturalness despite the reduced temporal sampling available for estimating the dense tracking field. FID and KID also degrade gradually, with no abrupt loss of reconstruction quality at the strongest 4× setting. Overall, the controlled trend across all metrics shows that ActDiff-VC can tolerate meaningful inaccuracies caused by large inter-frame displacement.

## A7.2 Motion Blur in the Tracker Input

This experiment evaluates whether AllTracker remains reliable when object boundaries and local textures are degraded by motion blur. Motion blur is applied exclusively to the video observed by the tracker, while the original sharp video remains unchanged for keyframe compression and evaluation. The tracker input is

$$x_t^{\text{trk}} = \mathcal{B}_b(x_t), \qquad b \in \{1, 11, 21\}, \tag{A2}$$

where $\mathcal{B}_b$ denotes the motion-blur operator with kernel size $b$, and $b = 1$ corresponds to the unmodified baseline. AllTracker estimates the dense tracking field from the blurred frames, after which the standard content-adaptive keyframe selection, budget-aware sparse trajectory selection, trajectory compression, and diffusion decoding procedures are applied.

Table A5 shows a small but monotonic reduction in reconstruction quality as the blur strength increases. With kernel size 11, LPIPS, FID, KID, and NIQE increase by 0.7%, 1.2%, 2.8%, and 1.1%, respectively. Even with the stronger kernel size of 21, the corresponding increases remain limited to 2.8%, 1.6%, 6.3%, and 2.0%.

These results indicate that the estimated dense tracking field remains sufficiently informative under moderate and strong blur at the evaluated levels. Although blur removes fine spatial details and weakens object boundaries, its effect is considerably smaller than that of temporal subsampling. This suggests that AllTracker can recover useful long-range correspondences from lower-frequency structural information and that the boundary keyframes and diffusion prior can compensate for modest local tracking inaccuracies.

## A7.3 Persistent Drift in Sparse Trajectory Conditioning

The previous experiments perturb the visual observations used to estimate the dense tracking field. In contrast, this experiment directly evaluates the sensitivity of the conditional diffusion decoder to inaccurate sparse trajectory conditioning. The dense tracking field and selected point set are first computed using the original video. Temporally correlated drift is then added to every trajectory in the selected sparse trajectory set $\mathcal{P}^{(k)}$.

For each selected point $q$, the drift is generated as

| Innovation standard deviation $\sigma$ | LPIPS ↓ | FID ↓ | KID ↓ | NIQE ↓ |
|:---:|:---:|:---:|:---:|:---:|
| 0 | 0.1708 | 29.51 | 0.0144 | 5.423 |
| 1 | 0.1754 | 30.13 | 0.0151 | 5.496 |
| 2 | 0.1805 | 30.42 | 0.0153 | 5.506 |
| 4 | 0.1846 | 30.87 | 0.0164 | 5.515 |

Table A6: **Analysis of robustness to persistent sparse-trajectory drift.** Even when temporally correlated localization errors are accumulated with $\rho = 0.9$, ActDiff-VC exhibits only gradual changes across the evaluated metrics, demonstrating substantial tolerance to inaccurate sparse trajectory conditioning even at the challenging setting of $\sigma = 4$ pixels. Lower is better for all metrics.

$$\mathbf{d}_1(q) = \mathbf{0}, \tag{A3}$$

$$\mathbf{d}_t(q) = \rho\,\mathbf{d}_{t-1}(q) + \boldsymbol{\epsilon}_t(q), \qquad \boldsymbol{\epsilon}_t(q) \sim \mathcal{N}\left(\mathbf{0}, \sigma^2\mathbf{I}_2\right), \tag{A4}$$

where $\rho = 0.9$ controls temporal persistence and $\sigma \in \{1, 2, 4\}$ pixels controls the innovation magnitude. The perturbed trajectory is

$$\widetilde{\mathbf{u}}_t(q) = \mathbf{u}_t(q) + \mathbf{d}_t(q). \tag{A5}$$

Because $\rho$ is close to one, the perturbation persists across neighboring frames and produces smooth trajectory wandering rather than independent frame-wise jitter. This behavior approximates accumulated localization error in long-range point tracking. The perturbed sparse trajectory set is compressed and passed to the decoder.

Table A6 demonstrates that ActDiff-VC remains robust even when substantial temporally persistent errors are introduced directly into the sparse trajectory set. At $\sigma = 1$, LPIPS, FID, KID, and NIQE increase by only 2.7%, 2.1%, 4.9%, and 1.3%, respectively. The degradation remains limited at $\sigma = 2$, with corresponding increases of 5.7%, 3.1%, 6.3%, and 1.5%. The strongest setting, $\sigma = 4$, constitutes a particularly challenging perturbation. Each trajectory receives Gaussian innovations with a standard deviation of 4 pixels per coordinate, and the high temporal correlation $\rho = 0.9$ causes these errors to persist and accumulate across frames. Despite this substantial trajectory drift, LPIPS and FID increase by only 8.1% and 4.6%, while NIQE changes by merely 1.7%. KID shows the largest relative increase of 13.9%, but its absolute change remains small, from 0.0144 to 0.0164.

The consistently small change in NIQE indicates that the reconstructed frames remain visually natural even when the transmitted motion condition deviates considerably from the original trajectories. The moderate changes in LPIPS, FID, and KID further show that the conditional diffusion decoder preserves perceptual similarity and distribution-level quality under substantial spatial inaccuracies. More importantly, reconstruction quality degrades smoothly with increasing $\sigma$, with no abrupt failure at any evaluated level. These results demonstrate that the generative prior and boundary-frame conditioning provide substantial tolerance to persistent errors in the sparse trajectory set.

## A8 Motion Encoder Construction

This section describes how the sparse trajectory set $\mathcal{P}$ is converted into the motion-conditioning latent used by the diffusion decoder. In principle, sparse trajectories could be encoded in different ways, for example through a learned trajectory encoder or another structured motion representation. In our implementation, since the decoder is built on DaS, we use a synthetic video representation that matches the DaS conditioning format. The transmitted side information consists of the sparse trajectory set $\mathcal{P}$, and the decoder deterministically converts $\mathcal{P}$ into this synthetic video representation.

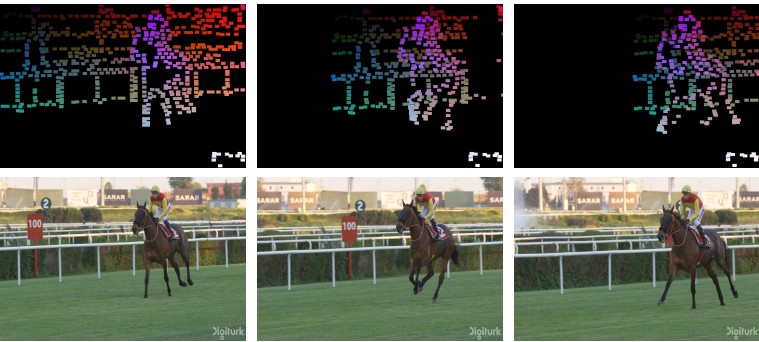

Figure A2: Visualization of the synthetic video representation used for motion conditioning. Given the sparse trajectory set $\mathcal{P}$, we generate a black-background RGB video in which each selected point is rendered as a small colored rectangle following its tracked trajectory. This synthetic video is then encoded by the pretrained VAE encoder $\mathcal{E}$ to obtain the tracking latent $z_{\mathrm{trk}}$ used by the diffusion decoder. The first row shows the generated synthetic video representation, while the second row shows the corresponding original video frames on which tracking is performed.

Let the sparse trajectory set for a segment of length $T$ be

$$\mathcal{P} = \left\{ \left( q_i, \{u_t(q_i)\}_{t=1}^{T} \right) \right\}_{i=1}^{N},$$

where $q_i \in \Omega$ denotes the initial location of the $i$-th selected point in the first frame, and $u_t(q_i) \in \mathbb{R}^2$ denotes its displacement at time $t$. From this set, we construct a synthetic RGB video

$$V_{\mathcal{P}} \in [0,1]^{T \times H \times W \times 3},$$

initialized with a black background. For each tracked point $q_i$ and each time step $t$, we compute its tracked position

$$p_{i,t} = q_i + u_t(q_i),$$

and render a small filled rectangle centered at $p_{i,t}$ on frame $t$. Each point is assigned a fixed color $c_i$ determined by its first-frame location, so that the same trajectory can be visually followed across time. The synthetic video representation is then mapped to the latent space by the pretrained VAE encoder $\mathcal{E}$. Accordingly, the tracking latent used by the diffusion model is

$$z_{\mathrm{trk}} = \mathcal{E}(V_{\mathcal{P}}) \in \mathbb{R}^{T_{\mathrm{lat}} \times h \times w \times 16}.$$

Equivalently, the trajectory-conditioning pathway denoted by $\mathcal{E}_{\mathrm{trk}}$ can be viewed as the composition

$$\mathcal{E}_{\mathrm{trk}}(\mathcal{P}) = \mathcal{E}(V_{\mathcal{P}}),$$

where the conversion from $\mathcal{P}$ to $V_{\mathcal{P}}$ is deterministic. This notation emphasizes that $\mathcal{E}_{\mathrm{trk}}$ is not an additional learned motion VAE used for compression; rather, it denotes the decoder-side procedure that transforms sparse trajectories into the latent motion-conditioning input required by the diffusion model.

Importantly, $V_{\mathcal{P}}$ is only an internal decoder representation. The bitstream contains the sparse trajectory set $\mathcal{P}$, not the synthetic video. Figure A2 illustrates this construction: the first row shows the synthetic video representation generated from the sparse trajectories, while the second row shows the corresponding original video frames on which tracking is performed.

## A9 Additional Distortion Metrics

We report PSNR and MS-SSIM to provide a distortion-oriented view of reconstruction fidelity. These metrics measure pixel-level and multi-scale structural similarity to the source video and are widely used in

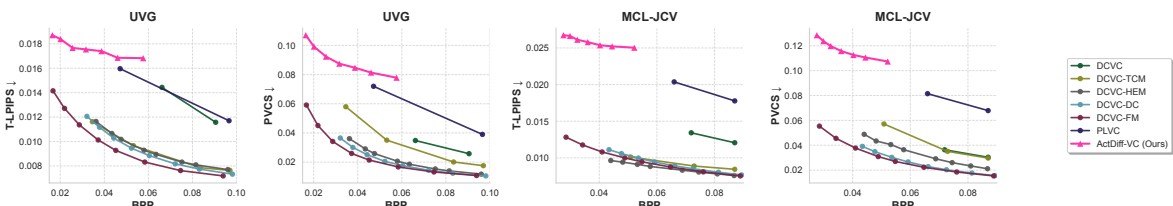

Figure A3: **Distortion-oriented evaluation on UVG and MCL-JCV.** We report PSNR and MS-SSIM as functions of bits per pixel (BPP). Higher values indicate better pixel-level or multi-scale structural fidelity to the source video. Distortion-optimized learned codecs achieve the strongest PSNR and MS-SSIM performance, while ActDiff-VC prioritizes perceptual realism at ultra-low bitrates through generative reconstruction from sparse conditioning.

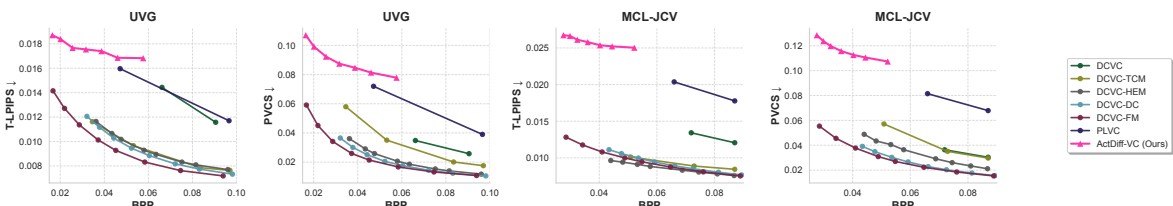

Figure A4: **Temporal evaluation on UVG and MCL-JCV.** We report T-LPIPS and PVCS as functions of bits per pixel (BPP). T-LPIPS compares the magnitude of frame-to-frame perceptual changes in the reconstructed and reference videos, while PVCS compares aligned clips using spatiotemporal I3D features. Lower values indicate closer agreement with the temporal evolution of the reference video.

conventional video compression evaluation. Figure A3 shows that distortion-optimized learned codecs achieve the highest PSNR and MS-SSIM on UVG and MCL-JCV, as expected from their predictive reconstruction designs and distortion-oriented training objectives. The GAN-based PLVC also achieves higher fidelity scores than ActDiff-VC, although its performance generally remains below that of the distortion-optimized codecs.

However, in the ultra-low-bitrate regime, PSNR and MS-SSIM do not fully capture perceived visual quality. At extremely low rates, distortion-optimized codecs may obtain higher PSNR and MS-SSIM while producing overly smooth reconstructions, whereas perceived quality can depend more strongly on realism, texture, and natural-image statistics. Therefore, PSNR and MS-SSIM should be interpreted as complementary fidelity measures rather than the primary indicators of visual quality in our target regime.

The lower PSNR and MS-SSIM values of ActDiff-VC reflect the expected perception–distortion trade-off of generative compression. ActDiff-VC prioritizes perceptual realism and distributional quality, as reflected by FID, KID, and NIQE, while sacrificing pixel-level and structural fidelity relative to distortion-optimized codecs.

## A10    Additional Temporal Metrics

We additionally evaluate temporal reconstruction quality using Temporal LPIPS (T-LPIPS) and Perceptual Video Clip Similarity (PVCS). We compute T-LPIPS using the LPIPS distance (Zhang et al., 2018) with an AlexNet backbone. Specifically, T-LPIPS measures the absolute difference between the perceptual change of consecutive reference frames and the corresponding change in the reconstructed video, averaged over all frame transitions. Lower values indicate closer agreement with the frame-to-frame perceptual changes of the source video. PVCS (Szeto & Corso, 2022) measures the distance between corresponding reference and reconstructed video clips using features extracted by a pretrained Inception-I3D network. We evaluate aligned, overlapping 10-frame clips using a temporal stride of one and average the normalized feature distances across multiple I3D layers and all clips. Lower PVCS values indicate closer agreement with the reference appearance and motion.

Figure A4 shows that ActDiff-VC obtains higher T-LPIPS and PVCS values than the competing codecs on both datasets. These results indicate that the temporal evolution generated by ActDiff-VC is less closely aligned with the exact frame-to-frame changes and spatiotemporal features of the reference video.

This gap partly reflects a limitation of generative reconstruction from sparse conditioning. ActDiff-VC reconstructs intermediate frames from compressed boundary keyframes and sparse point trajectories rather than directly coding the motion and residual information of every frame. Consequently, the generated motion, texture evolution, or local appearance changes may differ from the exact source sequence.

At the same time, T-LPIPS and PVCS are full-reference metrics and do not measure temporal coherence independently of reference fidelity. They penalize both undesirable temporal artifacts and visually coherent temporal changes that do not exactly reproduce the reference dynamics. Therefore, the results indicate weaker reference-aligned temporal fidelity, but do not by themselves imply a proportional reduction in perceived temporal coherence or overall viewing quality.

The human evaluation in Appendix A11 provides a complementary subjective assessment. In the pairwise experiment, participants were instructed to consider sharpness, naturalness, visible artifacts, temporal consistency, motion quality, and overall viewing pleasantness. ActDiff-VC was preferred over DCVC-FM in 62.59% of the comparisons, whereas DCVC-FM was preferred in 19.73%. Although this experiment does not isolate temporal consistency from the other aspects of video quality, it suggests that the gap in full-reference temporal metrics does not directly determine overall viewer preference.

Nevertheless, the T-LPIPS and PVCS results identify improved alignment with the source temporal dynamics as an important direction for future work, particularly through stronger temporal conditioning and improved consistency across independently reconstructed segments.

## A11 Human Evaluation

We conducted a human study to complement the objective perceptual metrics with subjective assessments of reconstructed video quality. A total of 147 participants completed two experiments: an absolute visual-quality rating task and a pairwise preference task.

Five source sequences were randomly selected from the UVG and MCL-JCV datasets. Each sequence was reconstructed using ActDiff-VC, DCVC-FM, and EVC-PDM at operating points of approximately 0.03 BPP, resulting in 15 reconstructed videos. Each video retained the frame rate and duration of its corresponding source sequence and was displayed at a resolution of $480 \times 720$. The original uncompressed videos were not shown; therefore, the study evaluated the perceived quality of the reconstructions rather than their explicit fidelity to a visible reference.

The study was completed using a laptop or desktop computer at normal screen brightness. Participants could replay each video before submitting their response, and the same participants completed both experiments. No practice trials were included.

Only the reconstructed video content was presented. Method names, filenames, URLs, and other identifying information were removed. For each participant, the ordering of the source sequences and reconstruction methods was randomized independently. In the pairwise experiment, the assignment of methods to Video A and Video B, as well as the order of the comparisons, was also randomized. Duplicate submissions were prevented. Incomplete or implausibly rapid submissions were designated for exclusion, although no responses met these exclusion criteria.

### A11.1 Absolute Visual-Quality Rating

In the first experiment, each participant independently rated all 15 anonymized reconstructed videos. Thus, every participant provided five ratings for each method, yielding 147 independent participant responses per video and 735 ratings per method. In total, 2,205 ratings were collected in this experiment.

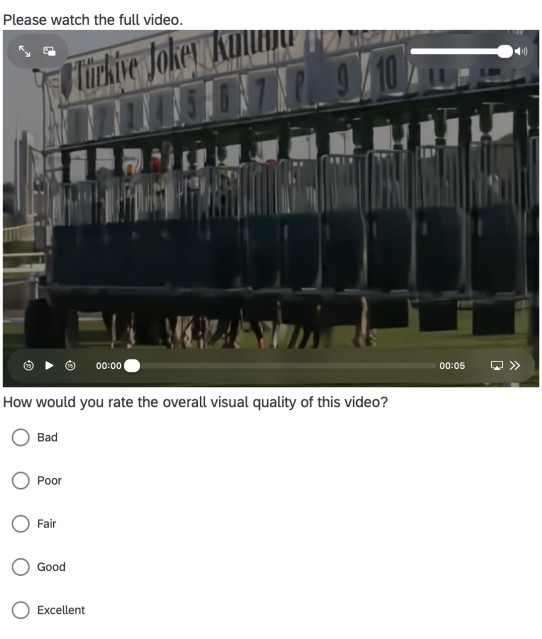

Figure A5: Interface used in the absolute visual-quality experiment. Participants viewed an anonymized reconstructed video and rated its perceived quality on a five-point scale from Bad (1) to Excellent (5).

| Method | MOS ± SD | Bad (1) | Poor (2) | Fair (3) | Good (4) | Excellent (5) |
|---|---|---|---|---|---|---|
| ActDiff-VC (Ours) | **3.31 ± 1.03** | 0.95% | 26.12% | 27.76% | 31.56% | 13.61% |
| DCVC-FM | 2.78 ± 0.94 | 14.69% | 13.88% | 50.07% | 21.36% | 0.00% |
| EVC-PDM | 1.51 ± 0.74 | 61.90% | 27.89% | 7.89% | 2.31% | 0.00% |

Table A7: Absolute visual-quality ratings at approximately 0.03 BPP. MOS is reported as mean ± standard deviation. Each method received 735 ratings from 147 participants over five videos. Percentages may not sum to exactly 100% because of rounding.

Participants evaluated each video according to their overall visual impression using the following five-point scale:

$$1 = \text{Bad}, \qquad 2 = \text{Poor}, \qquad 3 = \text{Fair}, \qquad 4 = \text{Good}, \qquad 5 = \text{Excellent}.$$

Participants were informed that there were no correct or incorrect responses and were asked to base their scores only on the perceived visual quality of the presented video. Figure A5 illustrates the rating interface.

Table A7 reports the mean opinion score (MOS), its standard deviation, and the distribution of responses across the five rating categories. ActDiff-VC achieved the highest perceived visual quality, obtaining a MOS of 3.31, compared with 2.78 for DCVC-FM and 1.51 for EVC-PDM. This corresponds to improvements of 0.53 and 1.80 MOS points, respectively. Moreover, 45.17% of the ActDiff-VC ratings were Good or Excellent, compared with 21.36% for DCVC-FM and only 2.31% for EVC-PDM. Conversely, only 0.95% of the ratings assigned to ActDiff-VC were Bad, whereas the corresponding proportions were 14.69% and 61.90% for DCVC-FM and EVC-PDM.

The average score of ActDiff-VC lies between Fair and Good rather than near the upper limit of the scale. This is consistent with the challenging ultra-low-bitrate operating point: at approximately 0.03 BPP, the available bit budget severely restricts the amount of visual information that can be preserved, and some loss of detail or visible reconstruction artifacts is therefore expected. Nevertheless, both the higher MOS

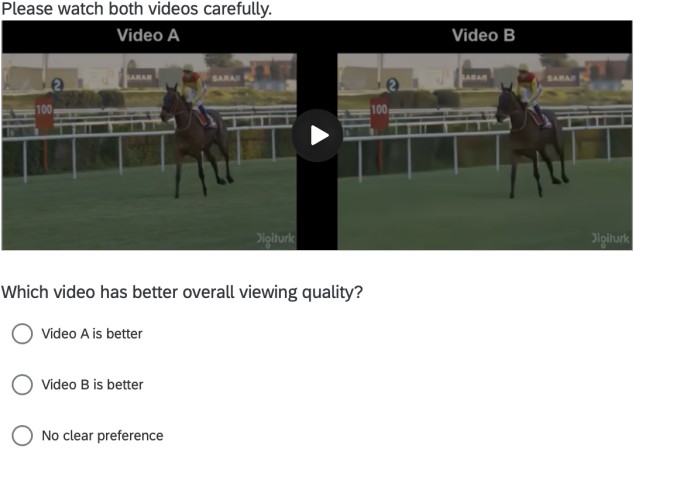

Figure A6: Interface used in the pairwise preference experiment. Participants viewed two anonymized reconstructions of the same source video and selected Video A, Video B, or No clear preference based on their overall visual impression.

| Baseline | Preference outcome (%) | | |
|---|---|---|---|
| | ActDiff-VC preferred | Baseline preferred | No clear preference |
| DCVC-FM | **62.59**% | 19.73% | 17.69% |
| EVC-PDM | **97.28**% | 0.00% | 2.72% |

Table A8: Pairwise preference results at approximately 0.03 BPP. Each entry reports the percentage of responses assigned to the corresponding preference outcome. Each comparison contains responses from 147 participants over five source videos.

and the shift toward the Good and Excellent categories show that ActDiff-VC provides substantially better perceived quality than the competing methods under the same severe rate constraint.

## A11.2 Pairwise Preference Evaluation

The second experiment evaluated relative visual preference. Participants were shown two anonymized reconstructions of the same source sequence, presented as Video A and Video B, and selected the reconstruction with the better overall viewing quality. They were instructed to consider sharpness, naturalness, visible artifacts, temporal consistency, motion quality, and overall viewing pleasantness. A *No clear preference* option was available when the difference was not sufficiently noticeable.

For each of the five source sequences, ActDiff-VC was compared separately with DCVC-FM and EVC-PDM. Consequently, each participant completed 10 pairwise comparisons: five against DCVC-FM and five against EVC-PDM. This produced 1,470 pairwise responses in total, with 735 responses for each baseline comparison. Figure A6 shows the pairwise evaluation interface.

Table A8 reports the percentage of responses assigned to each preference outcome. ActDiff-VC was preferred over DCVC-FM in 62.59% of the comparisons, whereas DCVC-FM was preferred in only 19.73%; the remaining 17.69% indicated no clear preference. This corresponds to a preference margin of 42.86 percentage points in favor of ActDiff-VC.

The comparison with EVC-PDM was even more decisive. Participants preferred ActDiff-VC in 97.28% of the comparisons, while no response favored EVC-PDM and only 2.72% indicated no clear preference. These

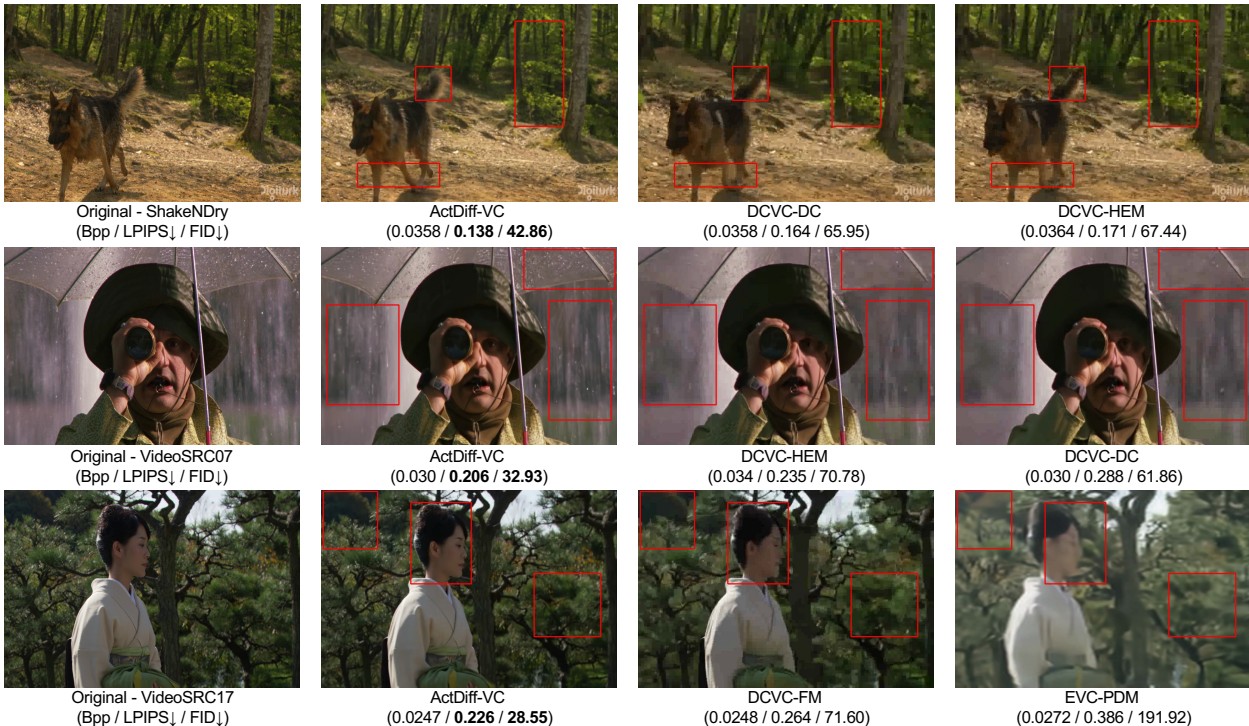

Figure A7: Additional qualitative comparison examples extending Figure 5 on representative sequences from the UVG dataset (ShakeNDry) and the MCL-JCV dataset (videoSRC07 and videoSRC17). For each example, we compare ActDiff-VC with competing methods at similar or higher bit rates. The numbers below each reconstruction report per-video BPP, LPIPS, and FID (lower is better). Red boxes highlight regions where competing methods exhibit stronger artifacts, blur, or structural degradation, while ActDiff-VC better preserves visually plausible textures and object boundaries.

pairwise results are consistent with the absolute quality ratings and show that the perceptual advantages of ActDiff-VC were clearly noticeable to participants at ultra-low bitrates.

Although the controlled perturbations evaluate larger apparent motion, motion blur, and persistent trajectory drift, they do not exhaust all real-world tracking failures.

## A12 Additional Visualization

Figure A7 provides additional qualitative comparisons that extend the results in Figure 5. Across these additional examples, the same trends are consistently observed: ActDiff-VC produces cleaner reconstructions with fewer artifacts and better preservation of fine structural details than competing methods. Additional examples of our content-adaptive keyframe selection, extending Figure 6, are shown in Figure A8 and Figure A9. These examples further confirm the robustness of our method under abrupt scene transitions.

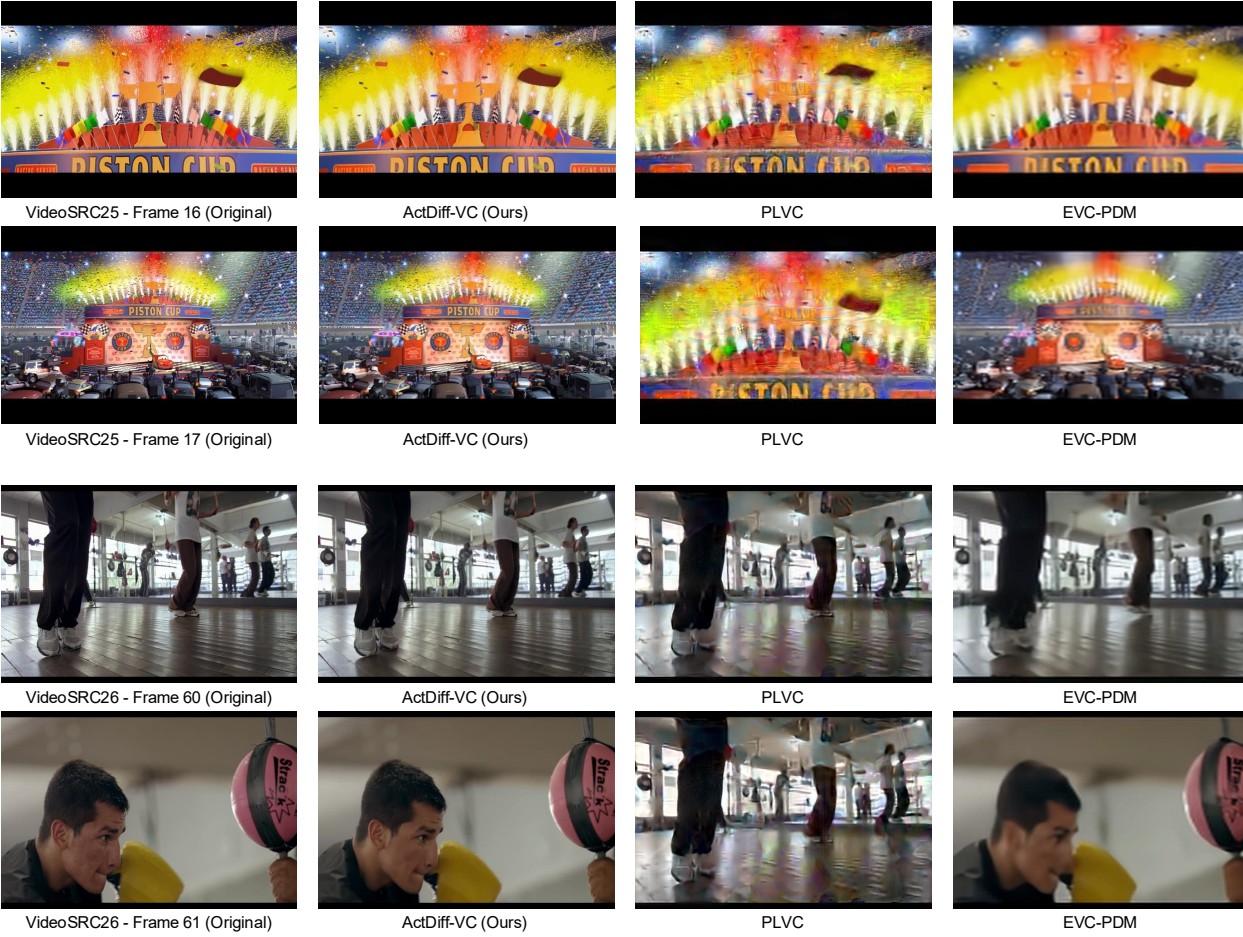

Figure A8: Additional visualizations of the proposed content-adaptive keyframe selection mechanism on MCL-JCV sequences videoSRC25 and videoSRC26. Similar to the examples in Figure 6, ActDiff-VC successfully adapts to abrupt scene changes by selecting new keyframes when needed, leading to higher-quality reconstructions after scene cuts. In contrast, PLVC often fails under its fixed key-frame schedule, while EVC-PDM requires substantially higher encoder-side computation and still produces weaker visual quality.

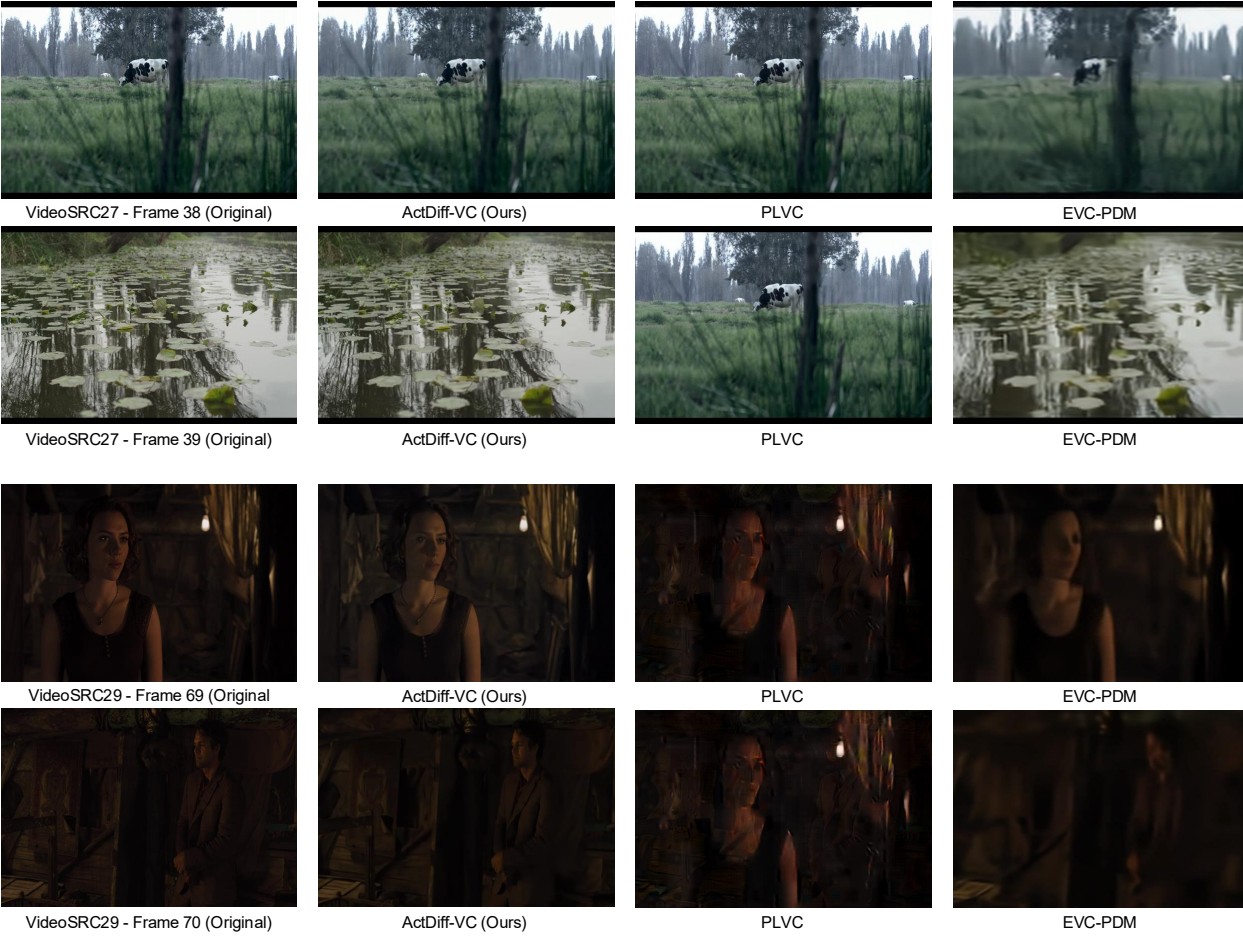

Figure A9: Additional visualizations of the proposed content-adaptive keyframe selection mechanism on MCL-JCV sequences videoSRC27 and videoSRC29. Similar to the examples in Figure 6, ActDiff-VC successfully adapts to abrupt scene changes by selecting new keyframes when needed, leading to higher-quality reconstructions after scene cuts. In contrast, PLVC often fails under its fixed key-frame schedule, while EVC-PDM requires substantially higher encoder-side computation and still produces weaker visual quality.

