# OpenReview forum: "Active Sampling for Ultra-Low-Bit-Rate Video Compression via Conditional Controlled Diffusion"
_TMLR — Under review for TMLR_

### Review · Reviewer_mVVK · 2026-06-12

**Summary Of Contributions:**

## Summary

This paper proposes ActDiff-VC, a diffusion-based video compression framework for the ultra-low-bitrate regime. The method adaptively partitions videos into variable-length segments, transmits compressed boundary keyframes and a sparse set of selected point trajectories, and uses a conditional video diffusion decoder to synthesize the intermediate frames. The paper further introduces content-adaptive keyframe selection and budget-aware sparse trajectory selection to reduce side-information cost while preserving perceptual reconstruction quality. Experiments on UVG and MCL-JCV show improved perceptual rate–distortion performance over several learned, GAN-based, and diffusion-based video compression baselines, particularly in terms of FID, KID, and NIQE at very low bitrates.



## Strengths

- **Clear motivation and problem setting:** The paper targets the ultra-low-bitrate video compression regime, where perceptual realism becomes more important than pixel-level distortion. This setting is well motivated and clearly distinguished from conventional learned video compression.

- **Reasonable and coherent framework:** The proposed combination of adaptive keyframe selection, sparse trajectory conditioning, and conditional diffusion decoding forms a coherent pipeline for reducing side information while leveraging a strong generative prior.



## Weaknesses

- **Limited fairness of comparison:** EVC-PDM is evaluated at 128×128 resolution, while most other methods are evaluated at 480×720. Although the authors attribute this to limitations of the released implementation, this resolution mismatch may affect the comparability of FID, KID, LPIPS, and NIQE.
- **Lack of fidelity and temporal consistency evaluation:** The paper mainly focuses on perceptual metrics, but diffusion-based reconstruction may generate visually realistic yet content-inaccurate frames. Important video compression metrics such as PSNR, MS-SSIM, temporal warping error, or LPIPS-t are missing, making it difficult to assess reconstruction faithfulness and temporal stability.
- **High decoding cost:** The proposed method still requires expensive diffusion-based decoding, with a reported decoding time of about 5528 ms/frame. Although this is faster than PLVC, it is much slower than many learned video codecs such as DCVC-FM, which limits the practical applicability of the method for general video compression scenarios.

**Audience:**

Yes

**Audience Explanation:**

The paper addresses an interesting and timely problem at the intersection of generative modeling and video compression. In particular, the use of sparse trajectory conditioning and conditional diffusion models for ultra-low-bitrate video reconstruction is likely to be of interest to researchers working on learned compression, perceptual reconstruction, and controllable video generation. Although I have concerns about the fairness of some comparisons and the completeness of the evaluation, the proposed framework and empirical observations could still be valuable to at least part of the TMLR audience.

**Claims And Evidence:**

No

**Claims Explanation:**

The submission provides useful experimental evidence on perceptual metrics such as FID, KID, NIQE, and LPIPS, and the reported results suggest that the proposed method can achieve strong perceptual quality in the ultra-low-bitrate regime. However, I do not think the evidence is fully convincing for all claims made in the paper. First, the comparison with EVC-PDM is conducted at a much lower resolution than the other baselines, which weakens the fairness and interpretability of the quantitative comparison. Second, the evaluation mainly focuses on perceptual metrics, while important aspects of video compression such as reconstruction fidelity and temporal consistency are not sufficiently evaluated. This is particularly important for diffusion-based codecs, which may generate visually plausible but content-inaccurate frames. Finally, the high diffusion decoding cost raises concerns about the practical applicability of the method. Overall, while the paper provides promising evidence for perceptual reconstruction quality, the current experiments do not fully support the broader claims about video compression performance and practicality.

**Requested Changes:**

- **Improve fairness of comparison (critical):** The authors should provide a more comparable evaluation against EVC-PDM, since it is currently evaluated at 128×128 while most other methods are evaluated at 480×720. If this is not possible due to implementation constraints, the paper should clearly discuss how this resolution mismatch may affect FID, KID, LPIPS, and NIQE, and avoid making overly strong conclusions from this comparison.

- **Add fidelity and temporal consistency evaluation (critical):** The authors should include additional metrics such as PSNR, MS-SSIM, temporal warping error, or LPIPS-t. This is important because diffusion-based reconstruction may produce visually plausible but content-inaccurate frames, and the current perceptual metrics alone do not sufficiently assess reconstruction faithfulness or temporal stability.

- **Discuss and mitigate decoding cost (strengthen):** The authors should provide a more detailed discussion of the practical implications of the high decoding cost, especially since ActDiff-VC requires about 5528 ms/frame for decoding. Additional experiments with fewer diffusion steps, accelerated sampling, or a clearer comparison of rate–quality–latency trade-offs would strengthen the paper.

---

> ### Author Response · Authors · 2026-07-15
> **Response to Reviewer mVVK (Part 1)**
>
> We sincerely thank the reviewer for the careful assessment of our work, for recognizing the clear motivation and coherent design of ActDiff-VC, and for identifying important limitations in the original evaluation. We agree that the resolution mismatch with EVC-PDM, the lack of distortion and temporal-fidelity metrics, and the computational cost of diffusion decoding required a more complete and transparent analysis.
>
> In response, we have substantially revised the manuscript. The new experiments and clarifications are highlighted in blue.
>
> ---
>
> ### 1. Fairness of the comparison with EVC-PDM
>
> We fully agree that directly including EVC-PDM in the original main comparison could obscure the effect of the resolution mismatch. The released EVC-PDM implementation supports only $128 \times 128$ inputs, whereas ActDiff-VC and the other methods in our main comparison are evaluated at $480 \times 720$.
>
> We have therefore made two changes.
>
> First, **EVC-PDM has been removed from the resolution-matched quantitative comparison in Figure 4**. The main comparison now contains only ActDiff-VC and baselines evaluated at the common resolution of $480 \times 720$. Consequently, the main quantitative claims concerning bitrate reduction and improvements in FID, KID, and NIQE do not rely on the comparison with EVC-PDM.
>
> Second, we added Appendix **A5, “Separate Comparison with EVC-PDM,”** which provides a more controlled supplemental evaluation at a common metric-computation resolution of $128 \times 128$. For this experiment:
>
> - EVC-PDM is evaluated using its native $128 \times 128$ reconstructions.
> - ActDiff-VC reconstructions and their corresponding source videos are resized to $128 \times 128$ before metric computation.
> - The same resizing procedure is applied to all frames and operating points.
> - We report LPIPS, FID, KID, and PSNR, covering perceptual, distributional, and pixel-level fidelity.
>
> We do not report NIQE or MS-SSIM in this comparison because their standard implementations require greater spatial support and do not provide reliable estimates at $128 \times 128$.
>
> | Dataset | ActDiff-VC BPP | EVC-PDM BPP | LPIPS reduction | FID reduction | KID reduction | PSNR improvement |
> |:--|--:|--:|--:|--:|--:|--:|
> | UVG | 0.0317 | 0.0332 | 52.4% | 84.7% | 91.4% | +8.7 dB |
> | MCL-JCV | 0.0360 | 0.0383 | 44.8% | 75.0% | 90.6% | +8.1 dB |
>
> At nearby operating points, ActDiff-VC performs substantially better across all four reported metrics on both datasets. Nevertheless, we emphasize that this is a **separate supplemental comparison at a common evaluation resolution**, rather than a native-resolution end-to-end codec comparison. EVC-PDM and ActDiff-VC still operate at different native spatial resolutions because of their released architectural constraints. We have therefore avoided using this experiment to support the principal resolution-matched claims in the main paper.

---

> > ### Author Response · Authors · 2026-07-15
> > **Response to Reviewer mVVK (Part 2)**
> >
> > ---
> >
> > ## 2. Reconstruction fidelity and temporal consistency
> >
> > We agree that perceptual metrics alone do not fully characterize a generative video codec. We have therefore added two new evaluation sections covering both reconstruction fidelity and reference-aligned temporal consistency.
> >
> > ### **Distortion-oriented fidelity**
> >
> > Appendix **A9, “Additional Distortion Metrics,”** reports PSNR and MS-SSIM on UVG and MCL-JCV for ActDiff-VC and the resolution-matched baselines. These metrics evaluate pixel-level and multi-scale structural fidelity to the source video.
> >
> > The new results show that distortion-optimized predictive codecs achieve higher PSNR and MS-SSIM than ActDiff-VC. We now report this result explicitly rather than suggesting uniform superiority across all quality dimensions. The lower distortion scores of ActDiff-VC reflect the expected perception–distortion trade-off of generative compression: ActDiff-VC prioritizes perceptual realism and distributional quality at ultra-low bitrates, while predictive codecs more closely reproduce the exact source pixels and structures.
> >
> > ### **Temporal consistency and source-aligned dynamics**
> >
> > Appendix **A10, “Additional Temporal Metrics,”** reports two complementary full-reference temporal metrics:
> >
> > - **T-LPIPS** measures the difference between the perceptual change of consecutive source frames and the corresponding change in the reconstructed video.
> > - **PVCS** compares aligned, overlapping 10-frame source and reconstructed clips using spatiotemporal features from a pretrained Inception-I3D video network.
> >
> > The results show that ActDiff-VC obtains higher T-LPIPS and PVCS values than the predictive codecs on both datasets. We now explicitly acknowledge that this indicates weaker alignment with the exact frame-to-frame evolution and spatiotemporal features of the source video.
> >
> > This behavior is consistent with the generative reconstruction setting. ActDiff-VC reconstructs intermediate frames from compressed boundary keyframes and sparse point trajectories rather than explicitly transmitting dense motion and residual information for every frame. Consequently, the generated motion, local texture evolution, or appearance changes may be visually plausible without exactly reproducing the source sequence.
> >
> > At the same time, T-LPIPS and PVCS are full-reference metrics: they penalize both undesirable temporal artifacts and coherent temporal changes that differ from the precise reference dynamics. They therefore measure reference-aligned temporal fidelity rather than perceptual temporal coherence alone.
> >
> > To provide complementary subjective evidence, Appendix A11 additionally reports a human evaluation with 147 participants. Participants were instructed to consider temporal consistency and motion quality together with sharpness, naturalness, artifacts, and overall viewing quality. Against the resolution-matched DCVC-FM baseline, ActDiff-VC was preferred in 62.59% of comparisons, while DCVC-FM was preferred in 19.73%.
> >
> > We have also revised the limitations section to explicitly identify weaker source-aligned temporal fidelity as a current limitation. Stronger temporal conditioning and improved consistency across independently reconstructed segments are now identified as important directions for future work.

---

> > > ### Author Response · Authors · 2026-07-15
> > > **Response to Reviewer mVVK (Part 3)**
> > >
> > > ## 3. Decoding cost and the quality–latency trade-off
> > >
> > > We agree that the original 50-step decoding time of 5528 ms per frame significantly limits practical deployment. In response, we added Appendix **A4, “Diffusion Step Trade-Off,”** which evaluates 50, 40, 30, 20, and 10 reverse-diffusion steps.
> > >
> > > For a controlled comparison, all configurations use the same encoded bitstreams, keyframe locations, sparse trajectory conditioning, sampler, bitrate, and inference parameters. Therefore, changing the number of diffusion steps affects only decoder-side computation.
> > >
> > > ### **UVG at 0.0186 BPP**
> > >
> > > | Diffusion steps | Decoding time (ms/frame) ↓ | LPIPS ↓ | FID ↓ | KID ↓ | NIQE ↓ |
> > > |--:|--:|--:|--:|--:|--:|
> > > | 50 | 5528 | 0.1708 | **29.51** | **0.0144** | 5.423 |
> > > | 40 | 4523 | 0.1716 | 29.88 | **0.0144** | **5.422** |
> > > | 30 | 3356 | 0.1717 | 30.20 | 0.0158 | 5.437 |
> > > | **20** | **2311** | **0.1694** | 30.25 | 0.0161 | 5.442 |
> > > | 10 | 1205 | 0.1844 | 36.19 | 0.0192 | 5.511 |
> > >
> > > ### **MCL-JCV at 0.038491 BPP**
> > >
> > > | Diffusion steps | Decoding time (ms/frame) ↓ | LPIPS ↓ | FID ↓ | KID ↓ | NIQE ↓ |
> > > |--:|--:|--:|--:|--:|--:|
> > > | 50 | 5528 | 0.1782 | 50.25 | **0.0198** | **5.144** |
> > > | 40 | 4523 | 0.1779 | 50.29 | 0.0203 | 5.150 |
> > > | 30 | 3356 | 0.1746 | 49.08 | 0.0199 | 5.173 |
> > > | **20** | **2311** | **0.1704** | **48.54** | 0.0207 | 5.151 |
> > > | 10 | 1205 | 0.1764 | 52.63 | 0.0241 | 5.162 |
> > >
> > > Reducing the sampling schedule from 50 to 20 steps lowers decoding time from 5528 to 2311 ms per frame, corresponding to a **58.2% reduction**. This improvement does not produce a consistent reduction in reconstruction quality. The 20-step setting achieves the best LPIPS on both datasets and the best FID on MCL-JCV, while remaining close to the best KID and NIQE results.
> > >
> > > By contrast, reducing the schedule to 10 steps is too aggressive: although it lowers latency to 1205 ms per frame, it noticeably degrades FID and KID on both datasets and LPIPS on UVG. We therefore adopt 20 steps as the default configuration in the revised manuscript.
> > >
> > > The updated end-to-end runtime analysis in Appendix A3 reports an encoding time of 109 ms per frame and a decoding time of 2311 ms per frame on a single NVIDIA A100 GPU at $480 \times 720$. Thus, ActDiff-VC provides a lightweight encoder while concentrating most computation at the generative decoder.
> > >
> > > We now clearly state in the abstract, introduction, and limitations that ActDiff-VC is not intended as a general-purpose real-time codec in its current form. Its computational profile is instead particularly suitable for ultra-low-bitrate applications such as cloud-assisted reconstruction, archival storage, and offline content distribution, where lightweight encoding, transmission efficiency, and perceptual reconstruction quality can justify greater decoder-side computation.
> > >
> > > ---
> > >
> > > Finally, we revised the manuscript to narrow and clarify our claims. We no longer imply that ActDiff-VC is uniformly superior across all reconstruction metrics or suitable for all video-compression applications. Our principal claim is specifically that ActDiff-VC provides favorable **perceptual rate–distortion performance in the ultra-low-bitrate regime**, particularly in FID, KID, NIQE, bitrate efficiency at matched perceptual quality, and subjective viewer preference, while exhibiting lower pixel-level fidelity, weaker reference-aligned temporal fidelity, and higher decoding cost than efficient predictive codecs.
> > >
> > > We sincerely thank the reviewer again. The concerns substantially strengthened the fairness, completeness, and transparency of our evaluation.

---

### Review · Reviewer_Yohf · 2026-06-15

**Summary Of Contributions:**

This paper presents ActDiff-VC, a generative video compression framework tailored for ultra-low-bitrate scenarios (≤ 0.05 bpp). The key motivation is that conventional distortion-oriented codecs often generate overly smooth and visually unsatisfactory results at such extreme compression rates. To address this challenge, the authors employ a conditional diffusion model as the decoder and introduce two lightweight yet effective encoder-side designs. Specifically, they propose a content-adaptive keyframe selection strategy that dynamically segments videos according to scene changes and motion complexity, as well as a budget-aware sparse trajectory selection mechanism that extracts compact motion cues from dense point tracking. These sparse trajectories are then used to guide the diffusion-based reconstruction process.

The primary limitation of the framework is its extremely high decoding cost. Because reconstruction relies on an iterative reverse diffusion process, the reported decoding time reaches 5528 ms per frame, making the system impractical for latency-sensitive or real-time applications. In addition, although the method demonstrates clear advantages in distribution-based perceptual metrics such as FID and KID, its performance is not uniformly superior across all quality measures. For example, at certain bitrate settings, competing methods such as DCVC-FM achieve better LPIPS scores, suggesting that the perceptual improvements may not always be reflected consistently across different evaluation criteria.

**Audience:**

Yes

**Audience Explanation:**

I believe this paper would be of considerable interest to the TMLR community. The integration of generative foundation models, particularly diffusion models, with neural compression has emerged as an active and rapidly evolving research direction. As the field increasingly shifts from distortion-oriented reconstruction toward perceptually optimized generation, understanding how to efficiently condition generative models under strict bitrate constraints becomes an important problem.

**Claims And Evidence:**

Yes

**Claims Explanation:**

Yes. Overall, the paper provides strong empirical evidence to support its main claims. The authors conduct extensive experiments on widely used video compression benchmarks, including UVG and MCL-JCV, and compare against a diverse set of competitive baselines covering predictive coding methods (e.g., the DCVC family), GAN-based approaches (PLVC), and diffusion-based compression frameworks (EVC-PDM).

**Requested Changes:**

- The paper should more explicitly discuss the significant computational asymmetry between encoding and decoding. While the encoder is efficient and practical, the decoding process remains extremely expensive, with a reported latency of more than five seconds per frame. This limitation has important implications for real-world deployment and should be acknowledged more clearly in both the abstract and introduction. Furthermore, the authors should explicitly identify the application scenarios for which such a trade-off is acceptable, such as cloud-based video archival, offline content distribution, or one-to-many streaming systems, rather than leaving this discussion primarily in the appendix.
- The robustness of the framework under tracking failures deserves further discussion. Since the diffusion decoder depends heavily on the trajectories provided by AllTracker, it would be valuable to understand how performance degrades when tracking quality deteriorates due to severe occlusions, motion blur, fast object movement, or challenging illumination conditions.
- Incorporating human subjective evaluations would significantly strengthen the empirical validation. Because the paper focuses on an ultra-low-bitrate setting where traditional distortion metrics become less informative, Mean Opinion Score (MOS) studies could provide direct evidence that the improvements observed in FID, KID, and NIQE translate into genuinely better viewing experiences. Such evaluations would complement the existing objective metrics and increase confidence in the practical significance of the results.

---

> ### Author Response · Authors · 2026-07-15
> **Response to Reviewer Yohf (Part 1)**
>
> We sincerely thank the reviewer for the careful assessment of our work, for recognizing the motivation and empirical evidence supporting ActDiff-VC, and for the constructive suggestions. We agree that decoding efficiency, robustness to tracking inaccuracies, and subjective video quality are particularly important for assessing a diffusion-based codec. In response, we have substantially revised the manuscript and added several new experiments, all highlighted in blue.
>
> ---
>
> ### 1. Computational asymmetry, decoding efficiency, and intended applications
>
> We fully agree with the reviewer that the computational asymmetry between the encoder and decoder should be stated clearly rather than discussed only in the appendix.
>
> In the revised manuscript, we have expanded Appendix **A3, “Encoding and Decoding Time Analysis,”** and explicitly acknowledge this asymmetry in the abstract, introduction, and limitations. All runtime measurements were obtained on a single NVIDIA A100 GPU at a resolution of $480 \times 720$.
>
> | Method | Encoding time (ms/frame) ↓ | Decoding time (ms/frame) ↓ |
> |:--|--:|--:|
> | DCVC | 1244 | 7137 |
> | DCVC-TCM | 353 | 129 |
> | DCVC-HEM | 360 | 81 |
> | DCVC-DC | 347 | 126 |
> | DCVC-FM | 140 | 116 |
> | PLVC | 40539 | 141659 |
> | **ActDiff-VC (Ours)** | **109** | 2311 |
>
> ActDiff-VC has the **fastest encoder among all compared methods**, requiring only \(109\) ms per frame. This end-to-end measurement includes dense point tracking, content-adaptive keyframe selection, HED-based sketch extraction, budget-aware sparse trajectory selection, keyframe compression, and entropy coding of the transmitted side information.
>
> We agree, however, that the decoder remains substantially slower than efficient feed-forward learned codecs because reconstruction requires iterative conditional diffusion. ActDiff-VC is therefore not intended for latency-sensitive or real-time decoding in its current form.
>
> To investigate whether this cost can be reduced without sacrificing perceptual quality, we added a new experiment in Appendix **A4, “Diffusion Step Trade-Off.”** We evaluate 50, 40, 30, 20, and 10 reverse-diffusion steps while keeping the bitstream, keyframe locations, sparse trajectory conditioning, bitrate, sampler, and all other inference settings fixed.
>
> | Diffusion steps | Decoding time (ms/frame) ↓ | UVG LPIPS ↓ | UVG FID ↓ | MCL-JCV LPIPS ↓ | MCL-JCV FID ↓ |
> |--:|--:|--:|--:|--:|--:|
> | 50 | 5528 | 0.1708 | **29.51** | 0.1782 | 50.25 |
> | 40 | 4523 | 0.1716 | 29.88 | 0.1779 | 50.29 |
> | 30 | 3356 | 0.1717 | 30.20 | 0.1746 | 49.08 |
> | **20** | **2311** | **0.1694** | 30.25 | **0.1704** | **48.54** |
> | 10 | 1205 | 0.1844 | 36.19 | 0.1764 | 52.63 |
>
> Reducing the sampling schedule from 50 to 20 steps decreases decoding latency from \(5528\) to \(2311\) ms per frame, a **58.2% reduction**. Importantly, the 20-step configuration achieves the best LPIPS on both UVG and MCL-JCV and the best FID on MCL-JCV, while remaining close to the best KID and NIQE values. By contrast, reducing the schedule to 10 steps causes clearer degradation, particularly in FID and KID. We therefore adopt 20 diffusion steps as the default configuration in the revised manuscript.
>
> These results clarify the intended trade-off: ActDiff-VC concentrates computation at the decoder to reconstruct perceptually realistic video from severely constrained transmitted information. The current system is best suited to **delay-tolerant ultra-low-bitrate applications**, including cloud-based or archival video reconstruction, offline content distribution, and one-to-many delivery settings in which content is encoded once and real-time receiver-side decoding is not required. We now state explicitly that the method is not currently appropriate for interactive or latency-sensitive applications.
>
> ---

---

> ### Author Response · Authors · 2026-07-15
> **Response to Reviewer Yohf (Part 2)**
>
> ## 2. Robustness to tracking failures
>
> We thank the reviewer for this important suggestion. In response, we added Appendix **A7, “Robustness to Tracking Failures,”** which evaluates how inaccuracies in the tracking signal propagate through the complete ActDiff-VC pipeline.
>
> We consider three complementary controlled perturbations:
>
> 1. **Temporal subsampling of the tracker input**, which increases the displacement between consecutive observations and approximates increasingly challenging fast-motion conditions.
> 2. **Motion blur applied exclusively to the tracker input**, which weakens object boundaries and removes local texture without modifying the keyframes, reconstruction targets, or evaluation videos.
> 3. **Temporally correlated drift added directly to the selected sparse trajectories**, which isolates the sensitivity of the conditional diffusion decoder to persistent localization errors.
>
> For temporal subsampling and motion blur, the perturbed dense tracking field is used for both content-adaptive keyframe selection and budget-aware sparse trajectory selection. These experiments therefore measure how tracking inaccuracies propagate through the complete encoder pipeline. For the trajectory-drift experiment, the dense tracking field, segment boundaries, and selected points are first computed from the original video; only the sparse trajectories passed to the decoder are perturbed.
>
> All other components and inference settings are kept fixed. The original unperturbed videos are used for keyframe compression, reconstruction targets, and metric evaluation. Results are averaged over all seven UVG sequences, and lower values are better for all metrics.
>
> ### **Temporal subsampling of the tracker input**
>
> For a subsampling factor s∈{1,2,4}, AllTracker observes every s-th frame. The estimated trajectories are then temporally interpolated to the original frame rate before content-adaptive keyframe selection, sparse trajectory selection, compression, and diffusion decoding.
>
> | Subsampling factor | LPIPS ↓ | FID ↓ | KID ↓ | NIQE ↓ |
> |:--:|--:|--:|--:|--:|
> | $1\times$ | 0.1708 | 29.51 | 0.0144 | 5.423 |
> | $2\times$ | 0.1811 | 31.03 | 0.0153 | 5.541 |
> | $4\times$ | 0.1881 | 33.45 | 0.0161 | 5.658 |
>
> Relative to the unmodified $1\times$ setting, $2\times$ subsampling increases LPIPS, FID, KID, and NIQE by 6.0%, 5.2%, 6.3%, and 2.2%, respectively. Under the more challenging $4\times$ setting, the corresponding increases are 10.1%, 13.4%, 11.8%, and 4.3%. The degradation is gradual, with no abrupt failure even when AllTracker receives substantially fewer temporal observations.
>
> ### **Motion blur in the tracker input**
>
> We next apply motion blur only to the frames observed by AllTracker. The original sharp frames remain unchanged for keyframe compression, reconstruction, and evaluation.
>
> | Blur kernel size | LPIPS ↓ | FID ↓ | KID ↓ | NIQE ↓ |
> |:--:|--:|--:|--:|--:|
> | 1 | 0.1708 | 29.51 | 0.0144 | 5.423 |
> | 11 | 0.1720 | 29.85 | 0.0148 | 5.483 |
> | 21 | 0.1755 | 29.98 | 0.0153 | 5.532 |
>
> With kernel size 11, LPIPS, FID, KID, and NIQE increase by only 0.7%, 1.2%, 2.8%, and 1.1%, respectively. Even with the stronger kernel size of 21, the corresponding increases remain limited to 2.8%, 1.6%, 6.3%, and 2.0%. These results show that the dense tracking field remains sufficiently informative even when local textures and object boundaries are substantially degraded.

---

> ### Author Response · Authors · 2026-07-15
> **Response to Reviewer Yohf (Part 3)**
>
> ### **Persistent drift in sparse trajectory conditioning**
>
> Finally, we directly perturb the selected sparse trajectories using temporally correlated drift. For each selected trajectory, we add temporally correlated drift according to $d_t=\rho d_{t-1}+\epsilon_t$, where $\epsilon_t$ is Gaussian noise with standard deviation $\sigma$. With $\rho=0.9$ and $\sigma \in \{1,2,4\}$ pixels. Because the error is temporally correlated, it produces persistent trajectory wandering rather than independent frame-wise jitter.
>
> | Innovation magnitude $\sigma$ | LPIPS $\downarrow$ | FID $\downarrow$ | KID $\downarrow$ | NIQE $\downarrow$ |
> |:--:|--:|--:|--:|--:|
> | 0 | 0.1708 | 29.51 | 0.0144 | 5.423 |
> | 1 | 0.1754 | 30.13 | 0.0151 | 5.496 |
> | 2 | 0.1805 | 30.42 | 0.0153 | 5.506 |
> | 4 | 0.1846 | 30.87 | 0.0164 | 5.515 |
>
>
> At $\sigma=1$, LPIPS, FID, KID, and NIQE increase by 2.7%, 2.1%, 4.9%, and 1.3%, respectively. At $\sigma=2$, the corresponding increases are 5.7%, 3.1%, 6.3%, and 1.5%. Even under the strongest setting, $\sigma=4$, LPIPS and FID increase by only 8.1% and 4.6%, while NIQE changes by 1.7%. KID increases from 0.0144 to 0.0164, corresponding to a relative increase of 13.9%.
>
> Across all three experiments, reconstruction quality degrades smoothly as the perturbation strength increases, without an abrupt failure mode. These results indicate that the compressed boundary keyframes, bidirectional boundary conditioning, and generative prior provide meaningful tolerance to imperfect tracking and inaccurate sparse trajectory conditioning.

---

> ### Author Response · Authors · 2026-07-15
> **Response to Reviewer Yohf (Part 4)**
>
> ### 3. Human subjective evaluation
>
> We greatly appreciate this suggestion and have added a comprehensive human study in Appendix **A11, “Human Evaluation.”**
>
> A total of 147 participants completed two complementary experiments:
>
> - an absolute visual-quality rating experiment using a five-point MOS scale; and
> - a pairwise preference experiment comparing ActDiff-VC separately against DCVC-FM and EVC-PDM.
>
> Five source sequences were randomly selected from UVG and MCL-JCV. Each sequence was reconstructed using the three methods at operating points of approximately 0.03 BPP. All videos were anonymized, method and sequence orderings were randomized independently, and participants were allowed to replay each video before answering.
>
> The absolute experiment collected 2205 ratings in total, corresponding to 735 ratings per method from 147 participants.
>
> | Method | MOS $\pm$ SD ↑ | Good or Excellent |
> |:--|--:|--:|
> | **ActDiff-VC (Ours)** | **3.31 $\pm$ 1.03** | **45.17%** |
> | DCVC-FM | 2.78 $\pm$ 0.94 | 21.36% |
> | EVC-PDM | 1.51 $\pm$ 0.74 | 2.31% |
>
> ActDiff-VC achieves the highest MOS, improving over DCVC-FM by 0.53 MOS points and over EVC-PDM by 1.80 points. Moreover, 45.17\% of ActDiff-VC ratings were Good or Excellent, compared with 21.36\% for DCVC-FM.
>
> The pairwise experiment collected 1470 responses. Participants were instructed to consider sharpness, naturalness, visible artifacts, temporal consistency, motion quality, and overall viewing pleasantness.
>
> | Baseline | ActDiff-VC preferred | Baseline preferred | No clear preference |
> |:--|--:|--:|--:|
> | DCVC-FM | **62.59%** | 19.73% | 17.69% |
> | EVC-PDM | **97.28%** | 0.00% | 2.72% |
>
> Against the resolution-matched DCVC-FM baseline, ActDiff-VC was preferred in 62.59\% of comparisons, while DCVC-FM was preferred in only 19.73\%. These results provide direct subjective evidence that the improvements in distributional and no-reference perceptual metrics translate into a noticeable improvement in overall viewing quality at ultra-low bitrates. We note that the original uncompressed videos were not displayed during this experiment. The study therefore evaluates the perceived quality of the reconstructions rather than their explicit fidelity to a visible reference.
>
> Finally, we have revised the discussion of our quantitative results to avoid implying uniform superiority across every metric. We now explicitly state that DCVC-FM achieves better LPIPS at some operating points, while the clearest advantages of ActDiff-VC occur in FID, KID, NIQE, bitrate efficiency at matched perceptual quality, and subjective viewer preference.
>
> We sincerely thank the reviewer again for these valuable suggestions. They led us to provide a more transparent account of the computational trade-off, a controlled evaluation of robustness to tracking inaccuracies, and direct subjective validation of reconstructed video quality.

---

### Review · Reviewer_HG4p · 2026-06-30

**Summary Of Contributions:**

The authors introduces ActDiff-VC,  a diffusion-based video compression framework for the ultra-low-bitrate regime. They used content-adaptive keyframe selection and budget-aware sparse trajectory selection. The key idea was to focus on perceived quality rather than pixel-level fidelity. Experiments shown ActDiff-VC remains superior among baseline methods.

# Strengths
S1) The paper is well driven and with good motivation.

# Weaknesses
W1) The evaluation section solely focused on perceptual metrics. As HED is involved in the method I doubt if it would have a slower runtime than baselines.The authors should conduct complexity/runtime analysis to report Encoder Time / Decoder time across the methods.

W2) Limited evaluations on video. The current evaluation section focused on image / keyframes, but the authors should also measure video perceptual metrics. Possible ways could be using Video-CLIP/VideoMAE or any pretrained video encoder to report cosine similarity or embedding distances, for measuring the temporal consistency / motion realism / clip-level semantic-perceptual similarity.

W3) The design feels heavily engineered and consisted a lot of module.

**Audience:**

Yes

**Audience Explanation:**

Yes, video compression

**Broader Impact Concerns:**

/

**Claims And Evidence:**

Yes

**Claims Explanation:**

Yes, sounds convincing to me

**Requested Changes:**

See Weaknesses.
W2 is pretty critical as the proposed method uses a generative video diffusion decoder, frame-level perceptual quality alone may hide temporal artifacts or motion inconsistency across frames.

---

> ### Author Response · Authors · 2026-07-15
> **Response to Reviewer HG4p (Part 1)**
>
> We sincerely thank the reviewer for the careful reading of our paper, the positive assessment of its motivation, and the constructive suggestions. We have substantially revised the manuscript in response to these comments. The newly added experiments and clarifications are highlighted in blue in the revised version.
>
> ### W1: Complexity and runtime analysis
>
> Thank you for highlighting the importance of computational efficiency. We apologize that the runtime analysis, which was included in Appendix A3 of the original submission, was not sufficiently visible.
>
> Appendix **A3, “Encoding and Decoding Time Analysis,”** reports end-to-end encoding and decoding latency for ActDiff-VC and all resolution-matched baselines. All methods were evaluated on a single NVIDIA A100 GPU at $480 \times 720$ resolution.
>
> | Method | Encoding time (ms/frame) ↓ | Decoding time (ms/frame) ↓ |
> |:--|--:|--:|
> | DCVC | 1244 | 7137 |
> | DCVC-TCM | 353 | 129 |
> | DCVC-HEM | 360 | 81 |
> | DCVC-DC | 347 | 126 |
> | DCVC-FM | 140 | 116 |
> | PLVC | 40539 | 141659 |
> | **ActDiff-VC (Ours)** | **109** | 2311 |
>
> Importantly, the reported ActDiff-VC encoding time is an **end-to-end measurement** that includes dense point tracking, content-adaptive keyframe selection, HED-based sketch extraction, budget-aware sparse trajectory selection, keyframe compression, and entropy coding of the transmitted side information. HED is applied only to the first frame of each content-adaptive segment to construct the sketch-weighted importance map; its cost is therefore already included in the reported encoder latency.
>
> Despite including the complete encoder pipeline, ActDiff-VC achieves the **lowest encoding latency among all compared methods**, requiring only 109 ms/frame, compared with 140 ms/frame for the next-fastest method, DCVC-FM, and 347–1244 ms/frame for the remaining learned codecs. Thus, the proposed content-adaptive segmentation and sparse trajectory selection do not introduce a large encoder-side computational burden.
>
> We agree that diffusion-based decoding remains more expensive than feed-forward learned codecs. To investigate and mitigate this cost, we have added a new experiment in Appendix **A4, “Diffusion Step Trade-Off.”** We evaluate the decoder using 50, 40, 30, 20, and 10 reverse-diffusion steps while keeping the encoded bitstreams, keyframe locations, sparse trajectory conditioning, bitrate, and all other inference settings fixed.
>
> Reducing the number of diffusion steps from 50 to 20 decreases decoding latency from 5528 to 2311 ms/frame, corresponding to a **58.2% reduction in decoding time**. This reduction does not cause a consistent loss in reconstruction quality: the 20-step configuration achieves the best LPIPS on both UVG and MCL-JCV and the best FID on MCL-JCV, while remaining close to the best KID and NIQE values. We therefore use 20 diffusion steps as the default configuration in the revised manuscript.
>
> These results demonstrate a favorable computational asymmetry: ActDiff-VC provides a highly efficient encoder while concentrating most of the computation at the generative decoder. We have also clarified that this trade-off is particularly relevant to ultra-low-bitrate settings in which perceptually realistic reconstruction from severely limited transmitted information is prioritized over real-time decoding.

---

> ### Author Response · Authors · 2026-07-15
> **Response to Reviewer HG4p (Part 2)**
>
> ### W2: Video-level perceptual and temporal evaluation
>
> We greatly appreciate this suggestion and agree that frame-level perceptual metrics alone are insufficient for evaluating a generative video codec.
>
> Following the spirit of the reviewer’s recommendation, we have added a new video-level evaluation in Appendix **A10, “Additional Temporal Metrics,”** using two complementary full-reference temporal metrics on both UVG and MCL-JCV:
>
> - **Temporal LPIPS (T-LPIPS):** compares the perceptual change between consecutive reconstructed frames with the corresponding change in the reference video. This evaluates whether the reconstructed sequence follows the frame-to-frame perceptual evolution of the source.
> - **Perceptual Video Clip Similarity (PVCS):** compares aligned, overlapping 10-frame reference and reconstructed clips using spatiotemporal features extracted by a pretrained Inception-I3D video network. This directly measures clip-level agreement in appearance and motion.
>
> The new results show that ActDiff-VC obtains higher T-LPIPS and PVCS values than the predictive learned codecs. We now explicitly acknowledge this result and its implication: although ActDiff-VC produces strong perceptual realism at ultra-low bitrates, its generated temporal evolution is less closely aligned with the exact frame-to-frame changes and spatiotemporal features of the reference video.
>
> This behavior is consistent with the design of generative compression. ActDiff-VC reconstructs intermediate frames from compressed boundary keyframes and a sparse trajectory set rather than explicitly transmitting dense motion and residual information for every frame. Consequently, the synthesized motion, local texture evolution, or fine appearance changes may not exactly reproduce those of the source video.
>
> We also clarify that T-LPIPS and PVCS are full-reference temporal-fidelity metrics: they penalize both undesirable temporal artifacts and temporally coherent changes that differ from the exact reference dynamics. Therefore, they measure reference-aligned temporal fidelity rather than perceived temporal coherence alone.
>
> To provide a complementary subjective assessment of complete reconstructed videos, we also added a human evaluation in Appendix **A11** involving 147 participants. Participants were explicitly instructed to consider sharpness, naturalness, visible artifacts, temporal consistency, motion quality, and overall viewing pleasantness. In the pairwise comparison against DCVC-FM:
>
> | Preference outcome | Percentage |
> |:--|--:|
> | **ActDiff-VC preferred** | **62.59%** |
> | DCVC-FM preferred | 19.73% |
> | No clear preference | 17.69% |
>
> ActDiff-VC also achieved a mean opinion score of 3.31, compared with 2.78 for DCVC-FM. Although this study does not isolate temporal consistency from the other dimensions of video quality, it indicates that the weaker full-reference temporal scores do not directly determine overall viewer preference.
>
> We have therefore expanded the evaluation from frame-level perceptual metrics to include both **objective clip-level temporal metrics and subjective complete-video evaluation**. We also explicitly identify improved alignment with the source temporal dynamics, particularly across independently reconstructed segments, as an important direction for future work.

---

> ### Author Response · Authors · 2026-07-15
> **Response to Reviewer HG4p (Part 3)**
>
> ### W3: The framework appears heavily engineered and contains many modules
>
> Thank you for raising this concern. We agree that the complete implementation diagram contains several processing blocks, and we have clarified the conceptual structure of the method in the revised manuscript.
>
> At the algorithmic level, ActDiff-VC is organized around only **two principal encoder-side mechanisms**, each addressing a necessary question in ultra-low-bitrate generative video compression:
>
> 1. **Content-adaptive keyframe selection:** determines *when* the current appearance anchor is no longer sufficiently informative and a new keyframe must be transmitted.
> 2. **Budget-aware sparse trajectory selection:** determines *which compact motion signals* should be transmitted between consecutive boundary keyframes.
>
> The conditional diffusion decoder then reconstructs the intermediate frames using the compressed boundary keyframes and the transmitted sparse trajectory set. Thus, the overall design directly follows the two sources of information required by the decoder: sparse appearance anchors and compact motion conditioning.
>
> The remaining blocks shown in the implementation diagram instantiate these two functions using standard or pretrained components. AllTracker estimates the dense tracking field, HED provides a fixed sketch-weighted importance map for trajectory selection, HiFiC compresses the boundary keyframes, and the pretrained DaS diffusion model performs generative reconstruction. These are not additional independently proposed or jointly trained subsystems.
>
> Moreover, Table 1 provides a controlled ablation study at a fixed bitrate, demonstrating that the principal design choices have distinct and measurable contributions:
>
> | Configuration or removed component | $\Delta$LPIPS ↓ | $\Delta$FID ↓ |
> |:--|--:|--:|
> | Full model: Adaptive GOP + sketch-weighted selection + bidirectional conditioning | 0.0000 | 0.0000 |
> | Remove bidirectional conditioning | +0.0321 | +1.3356 |
> | Remove sketch weighting | +0.0219 | +1.4193 |
> | Replace trajectory selection with uniform grid | +0.0246 | +5.6126 |
> | Replace trajectory selection with high-magnitude flow | +0.0458 | +12.4746 |
> | Fixed GOP + content-aware selection without sketch | +0.1323 | +24.7450 |
>
> The comparison between Adaptive and Fixed GOP under otherwise matched content-aware selection is particularly significant: changing to a Fixed GOP increases the degradation from +0.0219 to +0.1323 in LPIPS and from +1.4193 to +24.7450 in FID. Similarly, replacing the proposed sketch-weighted trajectory selection with uniform-grid or high-magnitude-flow sampling substantially reduces reconstruction quality. Removing bidirectional conditioning also causes measurable degradation.
>
> These results indicate that the components are not included as ad hoc engineering additions. Each serves a specific role required by the ultra-low-bitrate objective, and removing or simplifying these mechanisms leads to a clear deterioration in reconstruction quality.
>
> We sincerely thank the reviewer again for these suggestions. They motivated us to make the computational analysis more visible, add video-level temporal evaluation, and clarify the compact conceptual structure underlying the full implementation.

---

### Author Response · Authors · 2026-07-15
**Global Response: Summary of Rebuttal**

We sincerely thank all reviewers for their careful reading and constructive feedback. We appreciate their recognition of the clear motivation and timely problem setting of ultra-low-bitrate generative video compression, the coherent combination of adaptive keyframe selection, sparse trajectory conditioning, and diffusion-based reconstruction, and the strong empirical performance of ActDiff-VC on perceptual quality metrics.

In the revised manuscript, all additions and modifications are highlighted in blue. We strengthened the paper in four main directions:

**(i) Expanded computational analysis and clarified practical applicability.**
We made the computational asymmetry of ActDiff-VC explicit in the abstract, introduction, and limitations. We expanded the end-to-end runtime comparison and added a diffusion-step trade-off study using 50, 40, 30, 20, and 10 denoising steps. Reducing the sampling schedule from 50 to 20 steps lowers decoding latency from 5528 to 2311 ms/frame, a 58.2% reduction, while maintaining comparable perceptual quality. We now use 20 steps as the default and clearly state that the current method is intended for delay-tolerant ultra-low-bitrate applications, such as cloud-assisted reconstruction, archival storage, and offline content distribution, rather than real-time decoding.

**(ii) Improved the fairness and completeness of the evaluation.**
To address the resolution mismatch with EVC-PDM, we removed it from the main resolution-matched quantitative comparison. The main comparison now includes only methods evaluated at $480 \times 720$. We added a separate supplemental comparison with EVC-PDM at a common metric-computation resolution of $128 \times 128$ and clarified that the principal claims of the paper do not depend on this resolution-mismatched baseline. We also added distortion-oriented metrics, including PSNR and MS-SSIM, and full-reference temporal metrics, including T-LPIPS and PVCS, on both UVG and MCL-JCV.

**(iii) Added robustness and human-subjective evaluations.**
We added a new robustness analysis of tracking failures using three complementary perturbations: temporal subsampling of the tracker input, motion blur applied exclusively to the tracker input, and temporally correlated drift added directly to the transmitted sparse trajectories. The results show gradual degradation as tracking conditions become more challenging, without an abrupt failure mode. We also added a human evaluation with 147 participants, including five-point mean-opinion-score ratings and pairwise preference comparisons. ActDiff-VC achieved a MOS of 3.31 compared with 2.78 for the resolution-matched DCVC-FM baseline and was preferred over DCVC-FM in 62.59% of pairwise comparisons.

**(iv) Clarified the claims, limitations, and conceptual structure of the framework.**
We revised the quantitative discussion to avoid suggesting uniform superiority across all quality measures. The revised manuscript emphasizes the strongest advantages of ActDiff-VC in FID, KID, NIQE, bitrate efficiency at matched perceptual quality, and subjective preference, while explicitly acknowledging its higher decoding cost, lower distortion-oriented fidelity, and weaker reference-aligned temporal fidelity relative to predictive codecs. We also clarified that the framework is conceptually centered on two principal encoder-side mechanisms: content-adaptive keyframe selection and budget-aware sparse trajectory selection, with the remaining modules serving as standard or pretrained implementations of these functions.

We thank the reviewers again for their valuable suggestions, which substantially improved the fairness, completeness, transparency, and practical positioning of the manuscript.